



# PAPILA dataset: a regional emission inventory of reactive gases for South America based on the combination of local and global information

Paula Castesana[1, 2,3,*], Melisa Diaz Resquin[2,4,5,*], Nicolás Huneeus[5,6], Enrique Puliafito[1,7], Sabine Darras[8], Darío Gómez[2,4], Claire Granier[8,9], Mauricio Osses Alvarado[10], Néstor Rojas[11], and Laura Dawidowski[2,3]

[1]Consejo Nacional de Investigaciones Científicas y Técnicas, Buenos Aires, Argentina.
[2]Comisión Nacional de Energía Atómica, Gerencia Química, Buenos Aires, Argentina.
[3]Instituto de Investigación e Ingeniería Ambiental, Universidad Nacional de San Martín, Buenos Aires, Argentina.
[4]Facultad de Ingeniería, Universidad de Buenos Aires, Buenos Aires, Argentina.
[5]Center for Climate and Resilience Research $(CR)^2$, Santiago, Chile.
[6]Departamento de Geofísica, Facultad de Ciencias Físicas y Matemáticas, Universidad de Chile, Santiago, Chile
[7]Mendoza Regional Faculty – National Technological University (FRM-UTN), Mendoza, Argentina
[8]Laboratoire d'Aérologie, Université de Toulouse, CNRS, UPS, France
[9]NOAA Chemical Sciences Laboratory, Boulder, United States
[10]Departamento Ingeniería Mecánica, Universidad Técnica Federico Santa María (UTFSM), Santiago, Chile
[11]Air Quality Research Group, Universidad Nacional de Colombia, Bogotá, Colombia
[*]These authors contributed equally to this work.

**Correspondence:** Paula Castesana (pcastesana@unsam.edu.ar), Melisa Diaz Resquin (mdiazresquin@fi.uba.ar)

**Abstract.** The multidisciplinary project Prediction of Air Pollution in Latin America and the Caribbean (PAPILA) is dedicated to the development and implementation of an air quality analysis and forecasting system to assess pollution impacts on human health and economy. In this context, a comprehensive emission inventory for South America was developed on the basis of the existing data on the global dataset CAMS-GLOB-ANT v4.1 (developed by joining CEDS trends and EDGARv4.3.2 historical

data), enriching it with derived data from locally available emission inventories for Argentina, Chile and Colombia. This work presents the results of the first joint effort of South American researchers and European colleagues to generate regional maps of emissions, together with a methodological approach to continue incorporating information into future versions of the dataset. This version of the PAPILA dataset includes CO, $NO_x$, NMVOCs, $NH_3$ and $SO_2$ annual emissions from anthropogenic sources for the period 2014-2016, with a spatial resolution of 0.1° x 0.1° over a domain that covers 32°–120°W and 34°N–58°S.

PAPILA dataset is presented as netCDF4 files and is available in an open access data repository under a CC-BY 4 license: http://dx.doi.org/10.17632/btf2mz4fhf.2. A comparative assessment of PAPILA-CAMS datasets was carried out for (*i*) the South American region, (*ii*) the countries with local data (Argentina, Colombia and Chile), and (*iii*) downscaled emission maps for urban domains with different environmental and anthropogenic factors. Relevant differences were obtained both at country and urban level for all the compounds analysed. Among them, we found that when comparing total emissions of PAPILA

versus CAMS datasets at the national level, higher levels of $NO_x$ and considerably lower of the other species were obtained for Argentina, higher levels of $SO_2$ and lower of CO and $NO_x$ for Colombia, and considerably higher levels CO, NMVOCs



and $SO_2$ for Chile. These discrepancies are mainly related to the representativeness of the local practices in the local emissions estimates, to the improvements made in the spatial distribution of the locally estimated emissions, or both. Both datasets were evaluated relative to surface concentrations of CO and $NO_x$ by using them as input data to the WRF-Chem model for one of the

analysed domains, the Metropolitan Area of Buenos Aires, for summer and winter of 2015. For winter, PAPILA-based results had lower bias for CO and $NO_x$ concentrations, for which CAMS-based results tended to be underestimated. Both inventories exhibited similar performances for CO in summer, while PAPILA simulation outperformed $NO_x$ concentrations. These results highlight the importance of refining global inventories with local data to obtain accurate results with high-resolution air quality models.

## 1   Introduction

South America (SA) is a region of complex political and social contrasts, fluctuating economies and the highest inequality levels worldwide (The World Bank, 2019). Demographically, SA is a region with a growing population and an increasing trend towards urban agglomeration and the demand for goods and services (Huneeus et al., 2020a). From the energy standpoint, SA has significantly low coal consumption levels and higher share of hydroelectricity in comparison with other world regions

(IEA, 2020). The use of alternative fuels, such as biomass or waste, is usually not well covered by national statistics and therefore global information does not accurately represent the sectoral mix of fuels consumed in the different countries. Road transport in the region is characterized by a fleet older and in poorer operating and maintenance conditions than that circulating in developed countries. Moreover, the use of motorcycles has increased in the region, being of particular concern in some cities such as Lima and Bogotá (Romero et al., 2020; Ortegon-Sanchez and Oviedo Hernandez, 2016). In addition to diesel oil

and gasoline, different fuels are consumed for road transport in the region: compressed natural gas (CNG) covers a significant fraction of fuel use by passenger vehicles in Argentina, a high share of liquefied petroleum gas (LPG) is used in Peru, while pure ethanol and gasoline-ethanol blends are broadly used by flex fuel vehicles in Brazil (Belincanta et al., 2016). Adding to this diversity, legislation on sulfur content in fuels is very restrictive in some countries such as Chile and Colombia and much more flexible, particularly concerning diesel oil used by trucks and off-road vehicles, in others (Jorquera, 2002). With respect

to land use, SA is one of the least densely populated places in the world although it is highly urbanized (United Nations, 2015). This often implies poor or lacking information on the level and spatial distribution of some anthropogenic activities. This is the case, for example of the extended use in SA of wood and wood waste for cooking and for heating in colder zones of the region, e.g., southern Chile (Villalobos et al., 2017). Another relevant land use characteristic is the sustained trend of larger harvested land areas largely due to conversion from forests to agriculture lands (The World Bank, 2020), and partly as a consequence of

rising temperatures and changes in rainfall patterns, resulting in a shift of the agricultural border like in Argentina (Barros and Camilloni, 2016). Lastly, a unique feature of the region concerns hotspots of sulfur dioxide identified by the Ozone Monitoring Instrument (OMI) satellite sensor. For SA they are attributable mainly to volcanoes and the smelting of sulfides of copper and other metal ores in Chile and Peru, differing remarkably from other regions worldwide where these hotspots are mainly the responsibility of thermal power plants and oil and gas activities (Fioletov et al., 2016).



These regional particularities have direct consequences not only on the level and chemical profiles of the pollutants discharged to the atmosphere, but also on the specific locations where these emissions occur and on the population exposed to their environmental and health effects. Assessing the impact of atmospheric emissions as well as designing mitigation strategies, require reliable atmospheric emissions inventories (AEIs), which include spatially disaggregated emissions covering the entire region of interest in a transparent and consistent way in terms of emission sources and estimation methodologies (Kuenen

et al., 2014). There is a wide range of global AEIs covering SA for different species and periods that meet the mentioned requirements. Some of the AEIs worth mentioning include: the Emissions Database for Global Atmospheric Research (EDGAR) (Janssens-Maenhout et al., 2019; EDGAR, 2021), the Evaluating the Climate and Air Quality Impacts of Short-Lived Pollutants (ECLIPSE) (Stohl et al., 2015), the Community Emissions Data System (CEDS) (Hoesly et al., 2018), the integrated assessment model Greenhouse gas - Air pollution Interactions and Synergies (GAINS) (Klimont et al., 2017), or the Copernicus

Atmosphere Monitoring Service datasets (CAMS) (Granier et al., 2019).

Across the region, government efforts on AEIs are mainly focused on GHGs in line with the international commitments under the United Nations Convention on Climate Change (UNFCCC). The regional community of GHG inventory compilers has grown remarkably in the last two decades and in many cases has helped to improve the collection of activity data including specific areas of national statistics systems. In parallel, research groups in SA have built inventories of ozone precursors and

particles to be used as input data to air quality models. Links between several of these groups have recently been strengthened by the creation of a regional initiative focused on the construction of inventories of species not covered by governments in their reports to the UNFCCC (Huneeus et al., 2017, 2020a).

Completeness in terms of species, represented sectors and time series is a strength of global AEIs while locally developed inventories seldom fully cover all three aspects. On the other hand, although the most current versions of global AEIs accu-

rately reflect the emissions from sectors for which regional information is well documented in global statistics, they may miss some specificity and accuracy associated with local practices and technologies that is often better represented in local AEIs (Huneeus et al., 2020a). From this, it is plausible to assume that better emission estimates would be obtained by enriching the comprehensive global AEIs with locally generated information.

This work presents what to our knowledge constitutes the first AEIs from anthropogenic sources covering the entire SA

region, which combines in a proper and rigorous way local available information with a global database. For this purpose, the dataset CAMS-GLOB-ANT v4.1 (Granier et al., 2019), developed by joining CEDS trends and EDGAR v4.3.2 historical data, was used as a basis (hereinafter CAMS dataset), enriching it with locally developed inventories available in the bibliography until 2019, and selecting those with national coverage and with availability of data for the period and species of interest. The dataset presented in this work, hereinafter called PAPILA, focuses on the group of species known as reactive gases, given

their relevance in atmospheric chemistry as precursors of $O_3$ and $PM_{2.5}$: carbon monoxide (CO), nitrogen oxides ($NO_x$), non-methane volatile organic compounds (NMVOCs), ammonia ($NH_3$) and sulfur dioxide ($SO_2$) (Sharma et al., 2017). Due to the availability of data in the local AEIs and the completeness of the sectors represented, the 2014-2016 period was selected for this first version of the PAPILA dataset, including local information from Argentina (Puliafito et al., 2017; Castesana et al., 2018), Chile (Mazzeo et al., 2018; Gallardo et al., 2018) and Colombia (IDEAM, 2017). In addition, a comparison of the





performance of both AEIs (PAPILA and CAMS) is presented using near-surface CO and $NO_x$ mixing ratio simulated by the Weather Research and Forecasting-Chemistry (WRF-Chem) (Grell et al., 2005) at a high spatial resolution (3 km) against in situ observations made in Buenos Aires during February-March and August-September 2015.

This work was carried out within the framework of the Prediction of Air Pollution in Latin America and the Caribbean (PAPILA, 2020) and Emission Inventories in South America (EMISA, 2020) projects. PAPILA combines for the first time

an ensemble of state-of-the-art models, high-resolution emission inventories, space observations and surface measurements to provide real time forecasts and analysis of regional air pollution in the LAC region. Thus, an important aspect of the project is the development of appropriate and consistent surface emission inventories as input data for air quality models. EMISA initiative was created to lay the foundations for constructing robust and transparent inventories of the same set of species that have been consistently estimated across South American countries using the same methodological approach. Local information

on emissions was gathered from the countries that participate in the EMISA project: Argentina, Brazil, Chile, Colombia and Peru. Relevant research groups in Brazil and Peru have developed emission inventories for different cities (Policarpo et al., 2018; Dos Santos Lucon and Moutinho Dos Santos, 2005; Vivanco and Andrade, 2006; Romero et al., 2020; Dawidowski et al., 2014); however as far as we know they have have not developed inventories covering the entire countries for the species included in this study. For this reason, the information from these countries was not included in this first version of the combined

dataset. However, this work is expected to be the starting point for the preparation of comprehensive emission inventories in South America enriched with local information. For this purpose, we include a flow chart with the general methodology that we have applied in combining local information with a global dataset (see Figure 2).

The paper is organized as follows. Section 2 describes, for each country, the approach and sources of information used to develop the PAPILA dataset and also discusses the application of this inventory in an air quality model. Section 3 provides the

main differences of PAPILA and CAMS datasets for SA and other smaller domains and the results of the air quality simulations. Section 4 provides a description of the data availability, and finally Section 5 presents the main conclusions of this work.

## 2 Methods

### 2.1 PAPILA dataset overview

The PAPILA dataset is a collection of CO, $NO_x$, NMVOCs, $NH_3$ and $SO_2$ inventories of annual emissions from anthro-

pogenic sources in South America for the period 2014–2016. The inventories are presented as netCDF4 files, one for each species gridded with a spatial resolution of 0.1° x 0.1° covering the domain 32° W–120° W and 34° N–58° S. Each file contains 12 variables corresponding to the emissions in $Tg\,y^{-1}$ from the following categories, which are organized and denominated using the nomenclature given by CAMS: thermal power plants (ENE); residential and commercial combustion (RES); road transportation (TRO); non-road transportation (TNR); fugitive emissions (FEF); industries, including fuel consumption in man-

ufacturing industries and construction, refineries, industrial processes and solvent and other products use, (IND); agricultural soils (AGS); agriculture livestock (AGL); inland navigation, which includes domestic coastal, deep-sea and inland waterborne navigation, (SHP); international navigation (SHP-INT); waste (including solid waste, wastewater and incineration) (SWD);

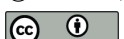



and the sum of all categories (SUM). This grouping of activities and sectors was carried out following the CAMS sectoral

disaggregation, except for the use of solvents, reported under IND in the PAPILA dataset. International navigation emissions

were taken entirely from the CAMS database. Agricultural fires were removed to allow the use of the inventories together

with fire products such as GFEDv3 (van der Werf et al., 2010), FINNv1 (Wiedinmyer et al., 2011) and GFASv1 (Kaiser et al.,

2012), avoiding double counting of these fires. It is worth mentioning that by "sum of all categories" we refer to all the sectors

included in PAPILA, both for the presentation of our results and for comparative purposes with CAMS. A broader description

of the activities contemplated under each category is presented in the Table A1 in the Annex A, together with the equivalences

with the IPCC 1996 reporting code.

The PAPILA dataset (Castesana et al., 2021) combines surface emissions from the comprehensive CAMS dataset with

local information of those countries that, at the time of development of this emission inventory, had emission estimates of the

mentioned species and covering the entire national territory: Argentina, Chile and Colombia. That information was collected

and assessed in terms of species, emission categories and spatial coverage, selecting the most appropriate and representative

data for each country, as described in the following subsections and summarized in Figure 1.

## 2.2 Designing and building the PAPILA dataset

Figure 2 presents a flow chart of the general methodology applied in combining local information with CAMS. Data from

Argentina, Chile and Colombia was assessed in terms of species availability. In addition, and as described in the next sub-

sections, the transparency on the methodology applied in emission estimates, and the representativeness and completeness of

emission sectors were revised in line with the CAMS emission reporting system. For those species and/or categories with

absence of local data, CAMS inventory was used to fill the gaps.

One of the challenges of combining different local inventories to a common regional database is bringing them to a single,

uniform and homogeneous grid. For this purpose, it was necessary to resolve all conflicts arising from cells shared by more

than one country or coastal cells. This problem was solved using the country and continent masks created at 0.1° resolution

(CIESIN and CIAT, 2005) that assign a unique value for each cell.

### 2.2.1 Argentina

Spatially disaggregated emission inventories for all the species included in this work are available for Argentina. They

cover all categories except SWD. Emissions from the following categories ENE, RES, TRO, TNR, FEF, IND and SHP were

taken from the GEAA inventory (Puliafito et al., 2017) which consists of a high-resolution (0.025° × 0.025°) inventory of

2014 annual emissions. For each category, GEAA covers: (*i*) for energy industries, the precise location of power plants, plus

fuel consumption by technology and by fuel of each utility; (*ii*) for residential and commercial, spatially distributed fuel

consumption estimated using energy use by province and census based population maps; (*iii*) for road transportation, fleet

composition, fuel consumption by refueling stations, geographically distributed considering road maps by type and distance

to the refueling stations; (*iv*) for off-road transportation, emissions from railways, with fuel consumption data, geographically

distributed with rail maps; (*v*) for fugitive emissions, including those from refining, storage, venting and flaring, and those







**Figure 1.** Local and global information on emissions and their spatial distribution included in the PAPILA dataset, by species, categories and countries. ENE: thermal power plants; RES: residential, commercial and other combustion; TR: road transportation; TRN: non-road transportation; FEF: fugitive emissions; IND: industrial process; AGS: agricultural soils; AGL: agriculture livestock; SHP: inland naviga- tion; SHP-INT: international navigation; SWD: wastes. CAMS refers to the global dataset CAMS-GLOB-ANT v4.1, and SA to the South American region. NO: Not occurring.

from distribution of oil products and natural gas, annual data from national statistics, spatially distributed with the exact location of the facilities; (*vi*) for inland navigation, fuel consumption, spatially distributed with the geographical identification of the berths routs and ports boundaries. The GEAA inventory has been updated for this work including emissions from IND,

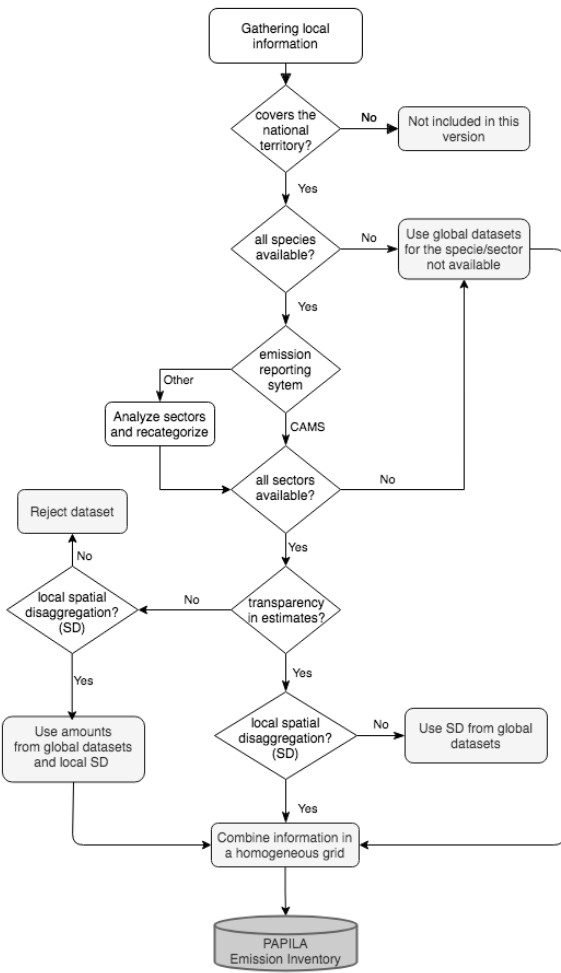

**Figure 2.** Flow Chart illustrating the general methodology applied to combine local information with the global dataset CAMS-GLOB-ANT v4.1.

which were not covered in the published version (Puliafito et al., 2017). With these changes for manufacturing industries, the

database considers fuel consumption by fuels, petroleum refining and emissions from the production process itself for the main industries, spatially distributed with the location of the main industries and distributing the rest as area sources in the whole territory. In all these categories the combustion of fossil and biomass fuels was considered. A different allocation of fugitive emissions from the distribution of oil products and natural gas (mainly consisting of NMVOCs) between the CAMS and the Argentinean inventory is worth mentioning. CAMS includes these emissions under the IND sector (see Table A1, Annex A)

while in the Argentinean inventory they are reported under FEF. This does not imply omission or double counting of emissions.



To construct the PAPILA inventory, this new version of GEAA was updated to 2015 and 2016 by applying CEDS trends by emission categories for Argentina and Chile and country-specific trends for Colombia. Final emissions were adapted to a homogeneous grid of 0.1° x 0.1°, and combined with agricultural local inventories described below and with the CAMS information on emissions from SWD.

Ammonia emissions from agricultural activities were taken from the 2000-2012 estimates by Castesana et al. (2018). The time series was updated to the period 2014–2016 by applying the methodology and local activity data sources detailed in the cited work. The rest of the studied species emitted from activities under AGL and AGS (*i.e.*, $NO_x$ and NMVOCs) were estimated according to the 2016 methodology of the European Monitoring and Evaluation Program (EMEP, 2017), following the general expression:

$$E_i = AD \cdot EF_i \tag{1}$$

where $E_i$ is the emission amount of the species $i$, AD is the activity data, and $EF_i$ represents the emission factor of the species $i$ related to that activity. Both the activity data and their spatial distribution were based on the previous work by Castesana et al. (2018, 2020), while the emission factors were those suggested by the EMEP according to the level of detail described for each activity in Table 1. These emissions have been estimated ad hoc to be included as part of the PAPILA dataset. Since

they have not been published thus far, interested readers may find a more complete description of the results in the Annex A, including resulting emissions from fertilizers, crop production and animal excreta (dairy and beef cattle, poultry, swine, sheep, goats, horses and other livestock). Consistent with the referenced studies cited above, emissions from managed excreta are reported as AGL, and those deposited in pasture during animal grazing are reported under AGS. Resulting inventories of annual emissions from agricultural activities spatially disaggregated at district level were taken to grids with a 0.1° x 0.1°

resolution.

### 2.2.2   Chile

   Chilean annual emissions were taken from the CR2-MMA dataset (CR2-MMA, 2018), based on the works of Gallardo et al. (2018) and Mazzeo et al. (2018). Such dataset is presented with a spatial resolution of 0.01° x 0.01°, and includes 2014 emissions of reactive gases, GHG and particles, reported under the following aggregation: industries (which includes

emissions from energy), urban and non-urban road transportation (which only includes CO and $NO_x$ for reactive gases), residential consumption, and agricultural and forest fires.

   Emission from industrial sources corresponds to the compilation of self-declared estimates by each facility to the Chilean Repository of Emissions and Pollutants Transport. Neither the methodology nor the emission factors used to estimate these emissions could be traced. However, given that the local methodology for $SO_2$ emission estimates is based on sulfur content in fuels and in mass-flow balances in copper production processes, which constitute the main $SO_2$ emitter activity in Chile

(González-Rojas et al., 2021), we have considered that the information on sulfur content that is handled locally is reliable, and included the spatially distributed emissions as estimated in Chile in our dataset. For the rest of the species, we decided to exclusively adopt the spatial distribution and the share of the locally reported emissions and distributed the CAMS estimates

**Table 1.** Level of detail of the EMEP 2016 methodology applied in estimating $NO_x$ and NMVOCs emissions from agricultural activities in Argentina.

| | Description of sources | EMEP 2016 approach | |
| --- | --- | --- | --- |
| | | $NO_x$ | NMVOCs |
| AGL | Manure management | | |
| | Dairy cattle, non-dairy cattle, | Tier 2 | |
| | and other livestock | | |
| AGS | Agricultural soils | | |
| | Inorganic N-fertilisers | Tier 1 | NO |
| | Manure in pasture (all | Tier 1* | Tier 2 |
| | livestock) | | |
| | Crops | NO | Tier 1 |

\* There is no EMEP Tier 2 method.

Emissions from grazing (deposited on pasture) are reported as agricultural soils. NO:

Not occurring. AGS: Agricultural soils. AGL: Agricultural livestock.

by weighing them on the CR2-MMA spatial distribution as follows:

$$E_{cell}(i,j,k) = E_{local}(i,j,k) \frac{\sum_{k=1}^{N} E_{\text{CAMS}}(i,j,k)}{\sum_{k=1}^{N} E_{local}(i,j,k)} \quad (2)$$

where $E_{cell}(i,j,k)$ are the emissions of species $i$ and sector $j$ assigned to the cell grid $k$, $N$ is the total number of grid cells covering the country, $E_{local}$ represents the emissions locally estimated and $E_{\text{CAMS}}$ the corresponding estimate from the global database.

Emission estimates from residential sources in the CR2-MMA dataset cover only firewood combustion for all species considered in our work (Mazzeo et al., 2018). According to local experts, one of the most relevant aspects of air quality in the coldest regions in southern Chile is the presence of CO, NMVOCs and particles from burning of firewood in households. Although this practice is included under the residential category of global inventories, the corresponding estimated emissions do not seem to be consistent with the magnitude of the air pollution situation observed at the local level. In addition, by downscaling global inventories in the Metropolitan Region of Santiago (MRS, Chile's central region) Huneeus et al. (2020a) found out that residential emissions were strongly overestimated in global databases and attributed this inaccuracy to the use of population density as a proxy for the spatial distribution. Residential emissions from firewood burning are less relevant in the more temperate northern areas where air pollution is mostly linked to emissions of $SO_2$ and particles from the metal industry (Huneeus et al., 2020b). From this and assuming that: (*i*) residential firewood burning is a predominant source in the southern region, (*ii*) in central and northern regions this work improves the representation of the diversity of sources and local practices for the other fuel combustion categories, we have decided to replace the residential emissions of the CAMS with those of the



CR2-MMA, at the risk of underestimating residential emissions in the central and northern regions by omitting those from fuels other than firewood.

Emissions of CO and $NO_x$ from urban and non-urban road transportation were added under the TRO category. Given that the local inventory reports ENE and IND (including use of solvent) emissions together and that insufficient information foro

spatial disaggregation was available, we to report ENE + IND under the IND sector for the case of Chile. Emissions taken from CR2-MMA° were extrapolated to 2015 and 2016 by applying CEDS trends (Hoesly et al., 2018) and projected to a 0.1° x 0.1° grid. Emissions from categories and species not estimated by the local inventory were taken from CAMS inventory.

### 2.2.3 Colombia

For the purpose of the PAPILA dataset, the only available information for Colombia was the emission estimates of CO, $NO_x$

and $SO_2$ at the national level from all the categories of interest, except agricultural soils. These estimates were developed for the Third National Communication of Colombia to the UNFCCC, covering the period 2010–2014 (IDEAM, 2017). Annual emissions from Colombia were extrapolated to 2015 and 2016 by applying linear regression forecast using the local time series and disaggregated using the spatial distribution of sources of the base inventory as follows:

$$E_{cell}(i,j,k) = E_{\text{CAMS}}(i,j,k) \frac{\sum\limits_{k=1}^{N} E_{local}(i,j,k)}{\sum\limits_{k=1}^{N} E_{\text{CAMS}}(i,j,k)} \tag{3}$$

where variables and indexes are those described in Eq. 2.

Although in this context the country estimates CO, $NO_x$ and $SO_2$ emissions from solid waste, wastewater and waste incineration, SWD emissions were taken from CAMS. The reason for this decision was that although the magnitude of the emissions was available, there was no information on their spatial distribution and it was not possible to applied the methodology described above, since CAMS considers zero SWD emissions for these species in Colombia.

### 2.3 Comparison of local and global datasets

A spatial analysis was performed following a similar approach to that by Trombetti et al. (2018) in their work on spatial intercomparison of top-down emission inventories in European urban areas, in which the analysis was made in terms of normalized emission values by group of categories and for different urban domains in order to become independent from emission levels and to show the relative contribution of a certain group of emission activities in different areas. However, we only need to

compare two inventories and are also interested in observing the differences in terms of magnitude, we therefore propose a comparison of normalized emissions by category and urban domain normalizing them with respect to those from the CAMS dataset, such as shown in Eq. 4. In this way, we were able to compare both datasets in relative terms and without losing





information on the shares of each group of categories and the differences in the emission levels of each dataset.

$$\forall i, \forall J : E^*_{i,J}(d, area) = \frac{\sum\limits_{k=1}^{N} E_{i,J}(d,k)}{\sum\limits_{k=1}^{N} E_{i,J}(\text{CAMS},k)} \tag{4}$$

where $E^*_{i,J}(d, area)$ and $E_{i,J}(d,k)$ are the normalized emissions and the emission levels, respectively, of the species $i$ and group of categories $J$ corresponding to the dataset d and the area (region, country or urban domain) covered by the total number $N$ of cell grids $k$.

For this analysis, we grouped categories as ENE + IND, RES, TRO, and "Other categories", and applied the analysis to (*i*) SA region, (*ii*) countries with local data (Argentina, Chile and Colombia), and (*iii*) urban domains from those countries

with local information on the spatial disaggregation of emissions. Urban domains were selected seeking to represent a wide variety of environmental and anthropogenic factors. In Chile, we have chosen three regions with different air quality concerns: Antofagasta (Northern region) with a strong presence of mining activity, Osorno (Southern region) a cold region where firewood burning dominates residential emissions, and the MRS (Central region) with a mix of emission sources and a strong presence of road transport. In Argentina, we have chosen three urban domains where relevant research groups are located, hoping that

this analysis would contribute to these activities. Those sites are: the Metropolitan Area of Buenos Aires (MABA) which is a coastal city and one of the main megacities in South America, Bahía Blanca (B. Blanca) which is a port city with an important industrial park, and Mendoza, one of the most important cities in the country that borders the Andes mountain range. A broader description of the studied areas is included in Table A3 and Figure A1 of the Annex A.

### 2.4    Dataset performance evaluation: case study in MABA

The performance of the PAPILA dataset in comparison with that of CAMS as input data to air quality models was assessed using the Weather Research and Forecasting Chemistry (WRF-Chem v4.1.2) regional model. The site chosen for this case study was the MABA, a megacity strongly influenced not only by mobile and residential sources, but also by the presence of four big thermal power plants, an important industrial park and an international port. Simulations were conducted using the model over three nested domains with a highest horizontal resolution of 3 km centered in MABA. Two time periods were

selected to cover summer (from 7 February to 5 March 2015) and winter (from 26 August to 17 September 2015) to assess the role of the emission estimates from the two inventories in the simulated air pollutant concentrations in the different seasons.

### 2.4.1    Model Description and Simulation Configuration

WRF-Chem is a fully coupled online chemistry transport model that simultaneously predicts weather and atmospheric composition (Grell et al., 2005). The simulations were done over 3 nested domains. The lowest resolution domain (d01) has

a grid size of 18 km x 18 km (51° W-78° W, 15° S-57° S), and the highest resolution domain (d03) has a 3 km x 3 km grid covering MABA and the surroundings. The coverage of the domains can be seen in Figure A2.





Lambert-conformal projections were used. The physical parameterizations adopted for the three domains were: a) Thompson scheme (Thompson et al., 2008) for Microphysics, b) Grell 3D scheme for cumulus parameterization, c) The Yonsei University scheme for boundary layer processes, d) MM5 similarity for surface processes , e) RRTMG scheme to compute long and

shortwave radiation. The chemistry in these simulations was modelled using the GOCART bulk aerosol scheme along with RADM2 for gas-phase chemistry for aerosol phase chemistry. The initial and boundary conditions were taken from the NCEP Final Operational Global Analysis data (FNL), available at a resolution of 1 degree every 6 hours (NOAA, 2000).

FINN Fire Database was used for fire emissions (Wiedinmyer et al., 2011), MEGAN Biogenic database for biogenic emissions (Guenther et al., 2006) and sea salt emissions from GOCART were also included.

Reported annual emissions in 2015 from the two inventories were processed to produce hourly-resolved emissions at resolution of each of the domains. For reactive gases we used the emission inventory presented in this article. The impact of aerosols emissions was also included in the analysis, enriching PAPILA database with EDGARv4.3.2 Global emission database for $PM_{2.5}$ and $PM_{10}$ and CAMSv4.1 for OC, BC and VOCs. Temporal emission patterns were computed by an iterative method, focusing on capturing daily and seasonal variations of CO, $NO_x$ and PM surface mixing ratios observed.

For ENE, TRO and RES monthly emission patterns were defined to breakdown total annual emissions into monthly fluxes (see Fig. A2). Emissions from other categories were evenly distributed throughout the year. RES monthly cycle was established from the reports on natural gas consumption reported in national statistics (ENARGAS, 2021) for residential and commercial activties in the MABA. This profile shows a maximum during winter linked to the increase in residential heating. Similarly, TRO monthly cycle was defined from the total fossil fuel consumption from the road transport reported in the statistics for

the entire country (Secretaría de Energía, 2021). For ENE, the same source of data was used to obtain monthly fossil fuel sales for thermal power plants in the Buenos Aires Province. Weekly cycles taken from PREP-CHEM (Freitas et al., 2011) were applied to the resulting total monthly emission fluxes. The hourly emission patterns were computed with the model by an iterative method in preliminary simulations, taken as seed the diurnal variations for transportation emissions reported by Wang et al. (2010) and focusing to capture the observed daily cycles and seasonal variation of CO, $NO_x$ and PM surface concen-

trations measured in the two monitoring stations, Parque Centenario and Córdoba. With this approach, the best configuration obtained with the simulations includes three hourly emission patterns: one related to diesel vehicle emissions, defined using PM observations, and the other associated with gasoline car emissions in winter and in summer, defined with the measured concentrations of CO and $NO_x$.

### 2.4.2   Model evaluation

Highest resolution model outputs using these two emission inventories were evaluated against CO and $NO_x$ ground-based observations from the available monitoring stations in MABA (See locations of the sites in Figure A2). Air pollutant data from the Environmental Protection Agency of Buenos Aires (APRA) include hourly measurements of $NO_x$ and CO in two sites, Cordoba (34.60º S, 58.39º W) and Parque Centenario (34.61º S, 58.44º W). Cordoba's site is located in a commercial area with high vehicular flow and very low incidence of stationary sources while Parque Centenario is located in a residential area next



to an arboreal space with medium vehicular flow and also very low incidence of stationary sources. As the air quality database is at hourly resolution, the model was also sampled at every hour.

The model evaluation was mainly focused on the effects of enriching the CAMS inventory with local inventories on the simulated air pollutant concentrations. For this purpose, median and percentiles for the entire period were evaluated. Also, mean daily concentrations were calculated to inspect whether model performance of both inventories was consistent and

satisfactory. Well-accepted statistical measures such as normalized mean bias (NMB), normalized mean gross error (NMGE) and the fraction of predictions within a factor of two (FAC2) were used (Wang et al., 2021). These statistical metrics were calculated using the following expressions:

$$NMB = \frac{1}{N} \frac{\sum_{k=1}^{N} S_k - O_k}{\sum_{k=1}^{N} O_k} \tag{5}$$

$$NMGE = \frac{1}{N} \frac{\sum_{k=1}^{N} |S_k - O_k|}{\sum_{k=1}^{N} O_k} \tag{6}$$

$$FAC2 : \frac{N(0.5 < \frac{S_k}{O_k} < 2)}{N} \tag{7}$$

where $S_k$ and $O_k$ are the simulated and observed hourly average concentrations respectively, and $N$ is the total number of observations. The model was sampled at each measuring location using grid interpolation and compared with the ground-based observations for the calculation of statistical performance metrics.

## 3 RESULTS AND DISCUSSION

Table 2 reports the resulting 2015 annual emissions of CO, $NO_x$, NMVOCs, $NH_3$ and $SO_2$ corresponding to the sum of all categories in the PAPILA dataset, for different domains: South America, Argentina, Chile, Bahía Blanca, MABA, Mendoza, Antofagasta, MRS and Osorno, in comparison with the corresponding emissions levels from CAMS. In addition, Figure 3 depicts the shares of each group of categories (ENE + IND, RES, TRO, and Others) to the total emissions of each species and for all the aforementioned domains. They are expressed in a normalized way with respect to the sum of all categories here

analysed in CAMS for each corresponding species and domain. In this way, the emissions of each species in CAMS correspond to the sum of the share of each group of categories, adding up to a total of 1.00. On the other hand, the shares of each sector in PAPILA can be compared with those in CAMS and add up to a total greater or less than 1.00 according to the differences in the sum of all the categories indicated in Table 2.

The spatial distribution of 2015 annual emissions of CO, $NO_x$, NMVOCs, $NH_3$ and $SO_2$ is shown in Figure 4 for the sum

of all categories. In addition, this figure includes maps with the differences between PAPILA and CAMS datasets for each



**Table 2.** Summary of annual emissions by domain for 2015 (Gg y$^{-1}$)

| 2015 (Gg y$^{-1}$) | CO | | NO$_x$ | | NMVOCs | | NH$_3$ | | SO$_2$ | |
|---|---|---|---|---|---|---|---|---|---|---|
| | PAPILA | CAMS | PAPILA | CAMS | PAPILA | CAMS | PAPILA | CAMS | PAPILA | CAMS |
| South America | 31225.2 | 32780.5 | 5791.0 | 5637.1 | 12186.0 | 11297.0 | 4885.5 | 5113.3 | 3248.4 | 3158.0 |
| Argentina | 2289.37 | 3740.3 | 901.2 | 660.2 | 548.7 | 1064.8 | 323.4 | 536.1 | 101.5 | 252.2 |
| Colombia | 1040.1 | 2343.4 | 328.3 | 367.7 | 798.0 | 798.0 | 395.1 | 395.1 | 177.5 | 146.3 |
| Chile | 3279.6 | 2080.5 | 307.3 | 355.0 | 1938.5 | 533.5 | 213.8 | 228.8 | 782.5 | 572.6 |
| MABA | 335.7 | 299.9 | 118.4 | 80.9 | 63.8 | 210.3 | 3.5 | 5.9 | 16.3 | 49.1 |
| Bahia Blanca | 9.6 | 21.5 | 16.8 | 7.3 | 2.6 | 7.3 | 0.3 | 0.4 | 3.7 | 16.1 |
| Mendoza | 39.9 | 49.1 | 16.8 | 10.5 | 8.0 | 25.6 | 1.0 | 0.6 | 1.2 | 5.5 |
| MRS | 112.2 | 345.4 | 35.0 | 31.8 | 30.7 | 136.6 | 1.8 | 9.1 | 14.7 | 38.9 |
| Antofagasta | 24.5 | 24.5 | 2.0 | 2.0 | 1.5 | 7.6 | 0.1 | 0.6 | 7.9 | 2.9 |
| Osorno | 136.7 | 19.5 | 4.4 | 1.6 | 88.7 | 3.6 | 1.5 | 1.4 | 0.4 | 0.3 |

species, depicting the differences in terms of intensity and location of emission sources. For comparative purposes, emissions from agricultural fires have been subtracted from the sum of all categories in CAMS database.

In what follows results are presented firstly by species, highlighting the most relevant aspects of the 2015 emission (section 3.1). Then, surface concentrations of CO and NO$_x$ obtained from the use of PAPILA dataset as input information in a chemical transport model in MABA, compared to those obtained using CAMS are presented (section 3.2). The section ends with an analysis of the local aspects that may have generated the difference between both emission databases (section 3.3).

### 3.1 Local-global comparison by species

#### 3.1.1 Carbon monoxide

Local estimates of CO for Argentina and Colombia presented lower CO annual emissions than those in CAMS, the largest differences occurred under the TRO category. Although the local estimates for TRO in Chile also showed significant smaller levels this difference was masked in the total CO national estimates by the larger emissions from the residential category, even after having omitted CO emissions from fuel combustion other than firewood. Lower CO PAPILA emissions in Argentina (-39%) and Colombia (-56%) were compensated by larger PAPILA emissions in Chile (58%) resulting in a difference with CAMS for South America of only -5%.

At the urban level, in the MABA domain PAPILA emission estimates resulted 12% higher than those from CAMS, this difference was mainly associated with higher local emissions from TRO and ENE + IND even when lower emissions resulted from RES. In the same way, Mendoza and B. Blanca exhibited lower CO total levels, mainly associated with differences in TRO and in less extent in RES. In B. Blanca, this difference masked the larger emissions by a factor of five in the local



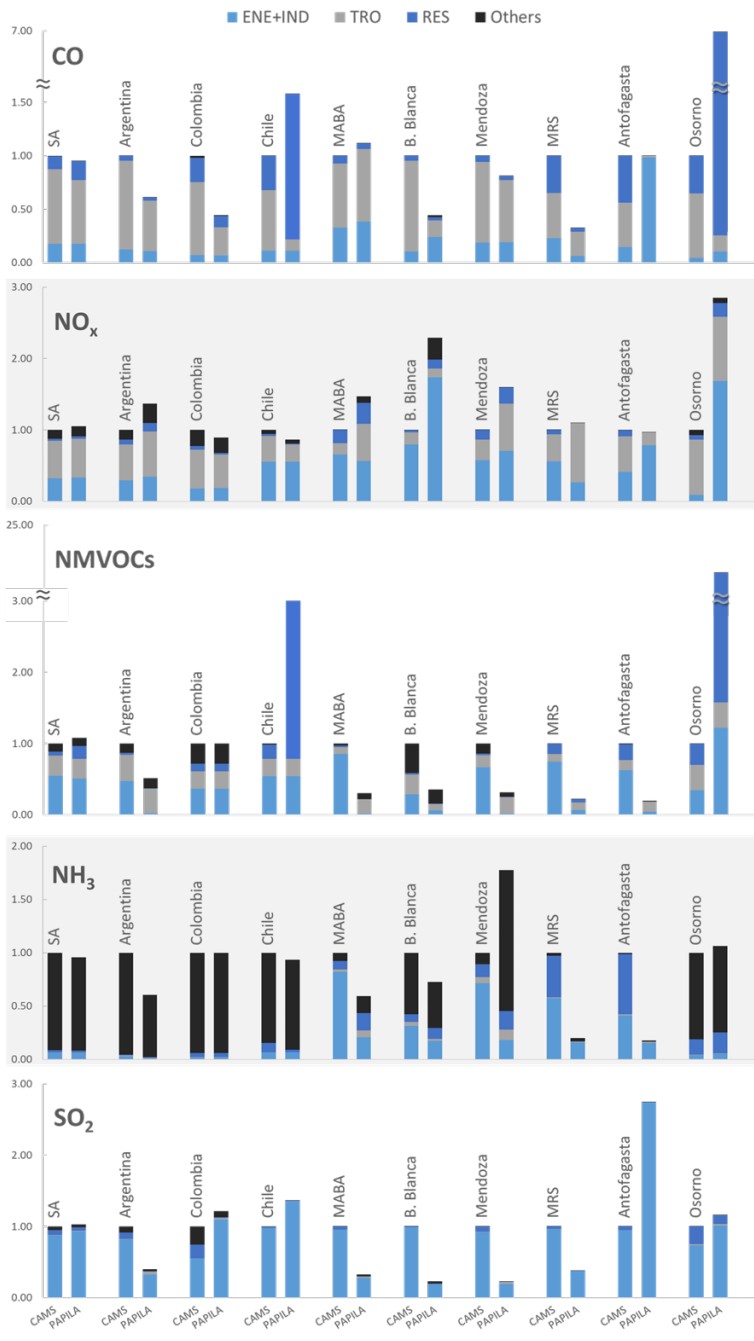

**Figure 3.** Normalized sectoral breakdown of PAPILA emissions compared with CAMS inventory by domain for 2015. Total CAMS emissions for each domain equal to 1.



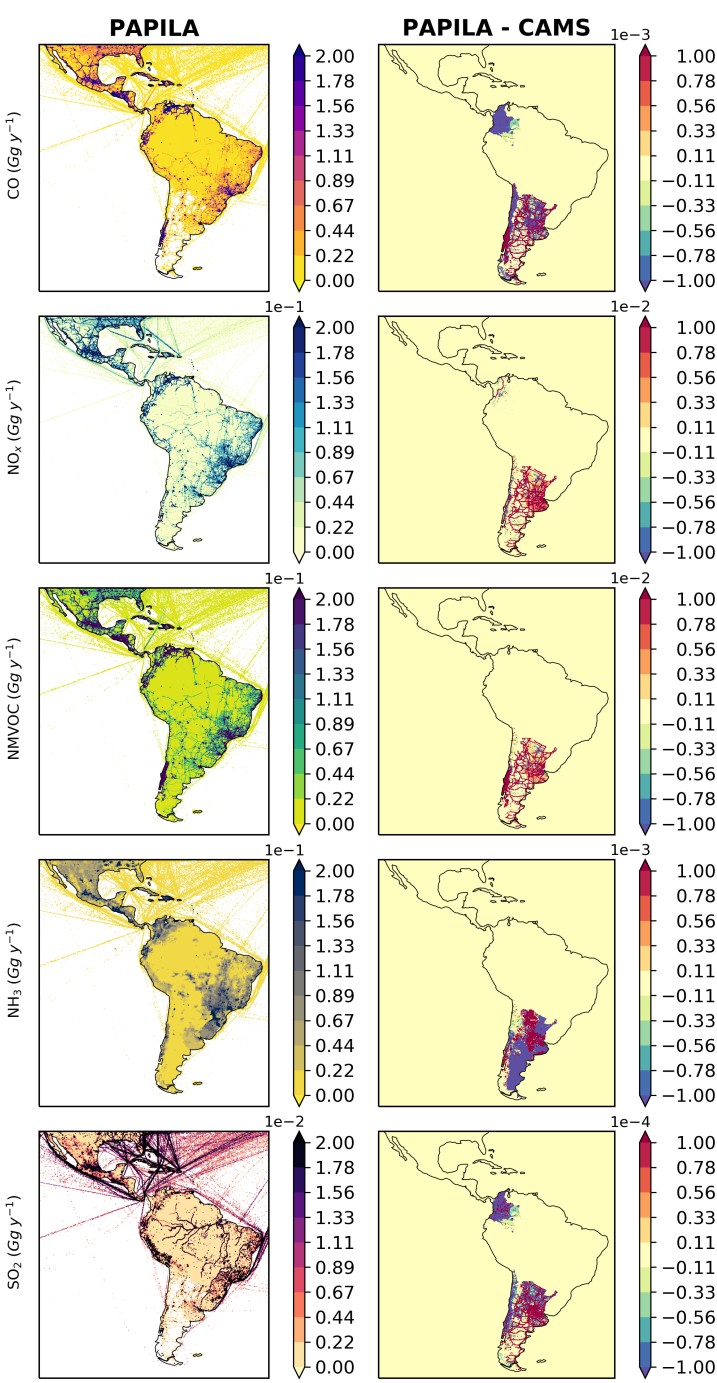

**Figure 4.** (left) Spatial distribution of 2015 PAPILA emissions (Gg y$^{-1}$) and (right) Difference between PAPILA and CAMS inventories by species in 2015.





estimates of ENE + IND with respect to the global dataset. By downscaling B. Blanca urban domain we identified the absence

of emissions from shipping activities (inland: SHP, and international: SHP-INT) in the global inventory. While emissions from SHP were estimated locally, estimates for SHP-INT were not available and therefore they were taken from global estimates. In this domain, international navigation is a concern since its port activity is almost as relevant as that of the international port of the Buenos Aires city (Ports, 2021). Although the absence of this source was not reflected in this comparative analysis, it is relevant to point it out that it could lead to underestimation of surface concentrations when modelling air quality in the region.

In the MRS, local estimates of total emissions were almost 70% lower than global ones, this difference is attributable to the two locally estimated categories that were included in this work (TRO, RES) but also to ENE + IND, categories for which we used a combination of the CAMS emission estimates with national information of location and emission shares. In Antofagasta, although total emissions levels from both datasets were similar, there were substantial differences in the contributions by category: emissions from ENE + IND are almost seven times larger in PAPILA than in CAMS, and while RES and TRO

emissions are negligible in the local estimates, according to global estimates they contribute to almost 90% of the domain's emissions. On the contrary, in Osorno local estimates for the sum of all categories were seven times larger than those in CAMS, emissions coming almost entirely from residential firewood burning.

### 3.1.2 Nitrogen oxides

Local estimates of national $NO_x$ emissions for Chile and Colombia were lower by 13% and 11%, respectively, than those

in CAMS. In both countries, the main responsible for these differences was the TRO category and to a lesser extent the lower emissions of RES, which in the case of Chile were due in part to the omission of the burning of fossil fuels in this category. In Colombia, local SHP estimates contributed to this difference, being partially offset by considerably higher emissions from TNR. Local estimates for Argentina resulted in higher total $NO_x$ emissions (37%) with very different sectoral contributions to this difference. The contributions by category (from highest to lowest) were TRO, ENE, AGS and RES, partially offset by

lower emissions from IND. All in all, $NO_x$ estimated emissions with local data for the whole SA region were 3% higher than those reported by CAMS.

As seen in Figure 3, all urban domains showed higher local emissions, except Antofagasta with a barely noticeable difference. In B. Blanca, the most relevant differences were the larger emissions in ENE + IND and SHP, category for which CAMS reported zero emissions while according to the local data it represented 12% of the domain's emissions, despite the omission

of the international port as a source of emission. Relevant larger emissions existed for TRO, ENE and RES in the MABA and Mendoza, together with significantly smaller emissions from IND. The larger emissions from other categories in the MABA and B. Blanca are mainly attributable to SHP, and although the impact of Other sectors on the total budget in these domains was negligible, local estimates showed considerably higher levels than CAMS for emissions from agricultural activities and FEF, category for which the global database attributed zero emissions in the three urban domains.

In the MRS, local emission estimates were slightly higher than those from CAMS, a difference mainly attributable to larger emissions from TRO, partially offset by smaller emissions in ENE + IND. In Antofagasta, by contrast, the larger emissions from ENE + IND sectors were almost completely masked by smaller TRO emissions. Local emissions from RES were strongly





underestimated in these two domains where according to CAMS data, RES is a minority emission source. In Osorno, local estimates of total emissions exceeded by a factor of almost three those of CAMS, being ENE + IND the sectors with the greatest

contribution to this difference and to a lesser extent TRO and RES. It is worth mentioning that the differences observed in ENE + IND in Chilean urban domains are exclusively associated with the local information on the spatial distribution and shares of $NO_x$ emissions, and not with a local estimate of the magnitudes.

### 3.1.3 Non-methane volatile organic compounds

Local estimates of NMVOCs emissions for Argentina were 48% lower, this difference is mainly attributable to IND (which

in this work includes solvent production and use). CAMS did not report NMVOCs emissions from agricultural activities (neither livestock nor soils) for any country in South America, while the local estimates for Argentina showed that 11% of the NMVOCs emissions came from these activities. PAPILA estimates of RES emissions in Chile (the only category locally estimated) exceeded those of CAMS by more than an order of magnitude, which was reflected in a total emission level three times larger for the country. The contribution of local Argentinean information, and that from RES in Chile, resulted in larger

regional emissions by 8%.

Local estimates showed important differences in total emissions for the three Argentinean domains (around 60-70% lower than CAMS), being IND the main contributor. Smaller emissions from FEF were observed in B. Blanca and Mendoza; while TRO contributed to these differences in the first domain and counteracted them in the second. In the MABA emissions from FEF and TRO were considerably larger than those in CAMS. Even when estimates from RES in PAPILA were around 80%

lower than those of CAMS they exhibited lesser impact on the differences between the two datasets and on the total emissions in each domain.

Local estimates for the MRS and Antofagasta were significantly smaller (around 80%) than the global ones, the difference is mainly attributable to the adopted local information on locations and emission shares for ENE + IND. On the contrary, local estimates for Osorno showed emissions more than 24 times larger than those of CAMS, almost exclusively attributed to the

incorporation of local information on firewood consumption in the RES category in cold areas of the country.

### 3.1.4 Ammonia

Similarly to NMVOCs, the only two countries with local data on $NH_3$ are Chile and Argentina. At national level, the inclusion of local information is reflected in differences of -7% of $NH_3$ emissions in Chile (only attributable to RES) and -40% in Argentina, where smaller emissions from AGS were the main responsible for that difference partially offset by larger

emissions from AGL. These two categories represent the main sources of $NH_3$ emissions in the country with a contribution of 72% from soils and 24% from livestock, according to the local estimates. Smaller emissions in local estimates of Argentina and Chile were reflected in a difference with CAMS of only -4% of total emissions in South America.

Although the impact of emissions from urban domains on the total levels of each country was negligible (around 1%), big differences were found at the category level between the two datasets. In Mendoza, local estimates resulted in larger

emissions by around 70% in total levels, mainly attributable to larger emissions from agricultural activities partly countered





by substantially lower emissions from IND. In B. Blanca, the difference in total levels was around -30% mainly attributable to ENE and AGS, while in the MABA the difference was around -40% where the main contribution to this difference was IND, partially offset by larger emissions from TRO, RES and other categories as SHP and AGL. Locally estimated emissions from RES were larger in the three urban domains.

Both Antofagasta and the MRS showed smaller total emissions by around 80% as a result of relocating ENE + IND emission sources, and replacing emissions from RES by the local inventory. In Osorno, slightly larger local estimates from RES and IND were observed. However, as in the case of urban domains in Argentina, the contribution of each domain to the total emissions in Chile was negligible.

### 3.1.5    Sulfur dioxide

Local estimates of $SO_2$ emissions in Argentina were 60% lower than those by CAMS for the sum of all categories, being IND, ENE and RES the main contributors to that difference (and SHP in a lesser extent), while TRO emissions were considerable higher (around eight times) in local estimates. For this country, these larger CAMS emissions were associated with the sulfur content adopted, mainly from solid fuels, since the national mineral coal has lower sulfur content ($370 \, \mathrm{kg} \, SO_2 \, \mathrm{Tj}^{-1}$) than those imported ($1100 \, \mathrm{kg} \, SO_2 \, \mathrm{Tj}^{-1}$), and because the national/imported ratio presented high variability between 2011 and

2015 (TCN, 2015). Colombia showed larger emissions from ENE, IND and TRO, partially offset by lower emissions from RES and SHP in the local estimates, and although negligible at the national level the emissions from FEF were significantly higher than in CAMS. As in Argentina, the sulfur content in the coal used was highly variable, due to the different sulfur levels that the country's coal fields present. In the same way as Colombia, Chile showed larger emissions from the sum of all categories as a consequence of the inclusion of local data in ENE + IND, differences mainly related with sulfur emissions from the

relevant copper mining activities that take place along the country, which were non fully covered by CAMS. The lower TRO emissions reported by CAMS for Argentina and Colombia seem to be related in part to the methodology used for projections, that assumes a sustained reduction in sulfur content from 2012 to 2015. Nevertheless, this reduction did not occur in any of the countries: while Colombia introduced prior to 2012 strong restrictions to fuel quality, in Argentina these restrictions for the fuels used by heavy duty trucks (the main emitters) did not take place. All this together results in larger emissions by only 3%

in the entire region.

Local estimates in the MABA showed smaller emissions by 67%, mainly associated with lower emissions from IND and RES (80-90%), the latter with less impact on the totals. This situation may be related to the fact that the proportion of S-emitting industries in the MABA is lower than in the rest of the country. Also with little impact, and offsetting these aforementioned differences, increases were observed in estimates from thermal power plants, inland navigation and transportation (TRO and

TRN). Both B. Blanca and Mendoza showed smaller emissions by 77%, mainly attributable to ENE in the first case and IND and RES in the second, where at the same time an increment in emissions from ENE was observed. Although the contribution of the TRO to the total emissions was minor in the urban domains of the country, the larger emissions estimated locally with respect to the CAMS is particularly noticeable.





**Table 3.** Summary of statistical metrics used for evaluating model performance to simulate surface CO and $NO_x$ concentrations in PAPILA and CAMS experiments for both sites. In Bold is highlighted the inventory that has the best performance for that metric in each period.

| | | Córdoba | | | | Parque Centenario | | | |
| | | Winter | | Summer | | Winter | | Summer | |
| | | PAPILA | CAMS | PAPILA | CAMS | PAPILA | CAMS | PAPILA | CAMS |
|---|---|---|---|---|---|---|---|---|---|
| CO | NMB | **-0.18** | -0.41 | **-0.15** | -0.39 | **0.02** | -0.39 | **-2E-3** | -0.28 |
| | NMGE | **0.43** | 0.51 | 0.58 | **0.56** | **0.39** | 0.50 | 0.60 | **0.52** |
| | FAC2 | **0.74** | 0.53 | **0.59** | 0.51 | **0.83** | 0.54 | 0.62 | 0.62 |
| $NO_x$ | NMB | **0.07** | -0.32 | **-0.08** | -0.41 | **-0.07** | -0.28 | 0.13 | **-0.12** |
| | NMGE | **0.58** | 0.63 | **0.52** | 0.69 | **0.62** | 0.63 | **0.61** | 0.63 |
| | FAC2 | **0.66** | 0.46 | **0.66** | 0.37 | **0.55** | 0.45 | **0.64** | 0.52 |

The MRS showed a difference in local estimates of around -62% mainly attributable to ENE + IND, while the result of
having included local estimates of emissions from these sectors in Antofagasta was reflected in total levels of $SO_2$ almost three
times larger than in CAMS. In these two domains, local estimates attributed to ENE + IND a contribution of more than 99% of
total emissions. In Osorno, although local emission estimates for RES were lower than CAMS, total emissions in the domain
resulted 16% larger than in global estimates as a consequence of the difference in emissions from ENE + IND.

## 3.2  Model evaluation and results

Table 3 summarizes the overall model performance of PAPILA and CAMS-based results for hourly CO and $NO_x$ concen-
trations. For the winter period, PAPILA-based results had lower normalized mean error than CAMS-based results; the negative
bias was larger for the CAMS emission run, exceeding in all cases more than 12% for both CO and $NO_x$ except for $NO_x$ in
Parque Centenario. FAC2 was also better in PAPILA simulation. Differences in the concentrations resulting from both runs
were consistent with those exhibited between the inventories. In terms of CO emissions, PAPILA dataset emissions were 12%
higher than CAMS being road transportation the main responsible followed by industry and residential sources. On the other
hand, $NO_x$ emissions were 46% higher in PAPILA, with significant discrepancies in emissions mainly from TRO followed
by ENE and IND. Emissions in the MABA are typically lower in summer because of decreasing traffic levels, null heating
requirements and lesser use of liquid fuels by thermal power plants, which burn almost exclusively natural gas during the
warmer period. Lower emissions levels coupled with favorable meteorological conditions for air pollutant dispersion conduct
to lower concentrations levels in summer. Thus, the goodness of the PAPILA-based results exhibited for winter were not that
apparent for summer. NMB was still negative with CAMS emissions, consistent with winter period results and better FAC2 for
PAPILA's run. Therefore, this highlights the importance of having accurate inventories especially for winter when the highest
emissions and worst dispersion conditions occur.



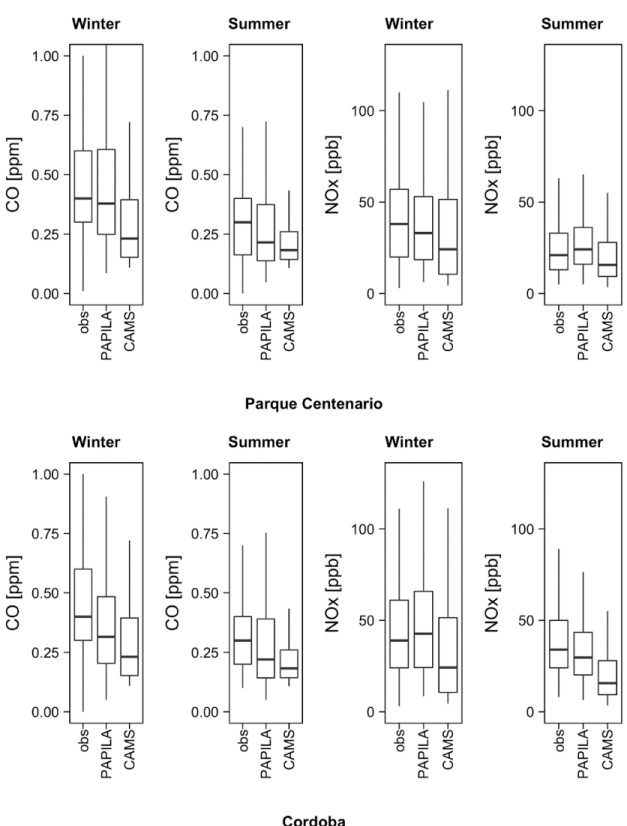

**Figure 5.** Box Plots of hourly CO and $NO_x$ in both sites for summer and winter simulations. The thick lines represent the medians for each period. The vertical hinges represent data points between the $25^{th}$ and $75^{th}$ percentiles. The whiskers represent data points between the $5^{th}$ and $95^{th}$ percentiles. CO concentrations are in ppm and $NO_x$ in ppb.

Figure 5 shows median and percentiles for hourly concentrations. The site Parque Centenario (residential) was better repro-
duced by the model with both inventories compared to Córdoba (commercial area with high traffic flow), especially in winter where the traffic flow in the city has a greater influence on the dynamics of these pollutants. However, we must also highlight that Cordoba monitoring station is located in a corner with high traffic flow and the sampling is at 2 m high. Therefore, the measurements may be capturing an overestimation of the real average activity in the area.

     Scatter plots of daily mean concentrations using PAPILA inventory (Figure 6) depicted a good agreement between obser-
vations and model, in winter more than in summer for CO and the other way around for $NO_x$. Concentrations estimated with CAMS inventory tend to be underestimated in all cases.

     All in all these results show a better agreement between observations and simulations using PAPILA dataset than CAMS, to represent surface concentrations of CO and $NO_x$ in the MABA. However, these emission improvements do not fully explain the underestimations of the model especially for CO concentrations with respect to the measured data.



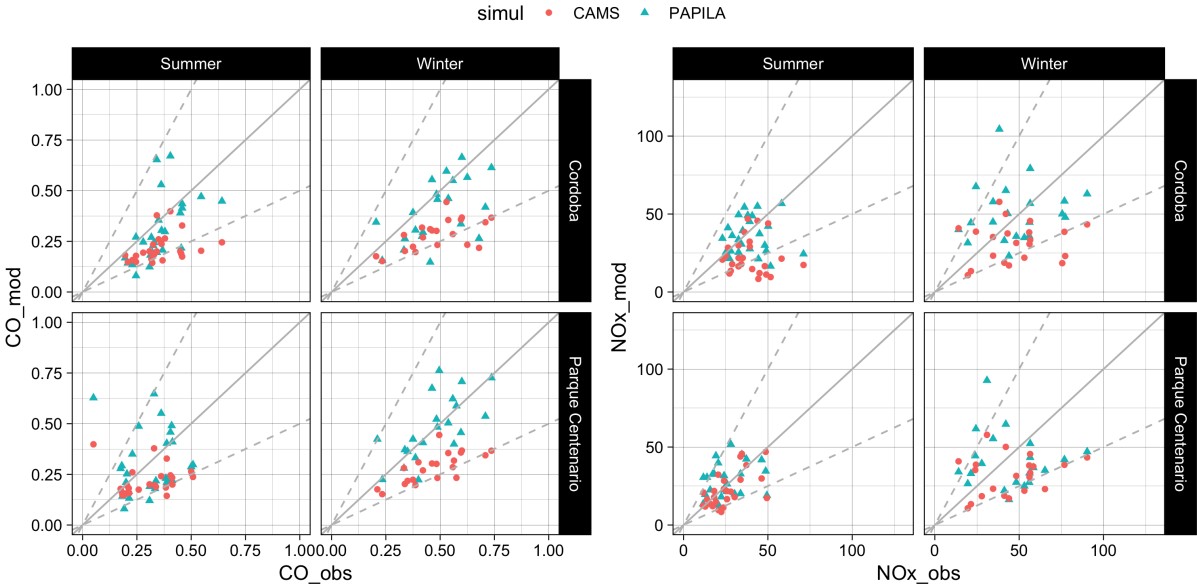

**Figure 6.** Scatter plots for observed and modeled daily concentrations for both sites. Values within the dotted lines represent the fraction of values that are within a factor of 2 of the observations (FAC2). CO concentrations are in ppm and $NO_x$ in ppb.

### 3.3 PAPILA-CAMS main differences

Our results show relevant differences between the PAPILA and CAMS datasets, both in terms of emission levels and their spatial distribution. They also served to exemplify the goodness of the PAPILA-based simulated surface mixing ratios in the city where concentrations were analysed.

The reasons behind the observed differences are diverse and are mainly linked to the activity data and to the methodologies applied to estimate and spatially distribute the emissions. Given the limited availability of local data, the emission factors for the compounds covered in this work were mostly based on default values in both the global and local datasets considered. In general, it is observed that the CAMS-GLOB-ANTv4.1 applies linear trends on the emissions of the aggregated categories, based on the estimates of the EDGARv4.3.2 of the year 2012. This is different from what has been done in this work, since for the three countries 2014 was taken as the base year, and while the same methodology than that used by CAMS was applied to extrapolate to 2015 and 2016 for Chile and some categories for Argentina, locally estimated trends were applied for Colombia. In this way, local inventories are not only based on region-specific information, but also the extrapolation of a shorter time period reduces the uncertainties associated with the activity data strongly linked to short-term variations and technological changes.

For those activities related to fuel consumption, local inventories used the information reported in the national energy balance and other national energy statistics while global inventories are based on the information reported by the countries to the International Energy Agency, which is consistent but not exactly the same than that reported in the national statistics since





the IEA processes the information received (IEA, 2020). Moreover, although these statistics adequately represent the national energy balances, it is worth pointing out the lack of specificity in terms of spatial disaggregation. A relevant aspect of the fuel consumption patterns for power generation in the three countries analysed, is their inter-annual and inter-regional variation,

which in turn are strongly correlated with hydraulic availability not captured by the extrapolations. In addition, in order that electricity supply matches demand, some short-term technological changes are often resorted to. For example, the incorporation of diesel-based motor generators located in different urban centers in Argentina such as the MABA in 2014 (CAMESA, 2021). Although the diesel consumed in these generators was reflected in international statistics, they did not distinguish between the gas oil used for this purpose and that used by combined gas cycles and could not reflect their location and operating regimes.

Another relevant aspect of national and therefore international statistics is the lack of reliable information on firewood consumption, widely used in rural and urban areas of the cold regions of Chile. This fact also impacts on the correct representation of the replacement of firewood by LPG or Natural Gas that took place in Argentina in the last decade, due to higher production of non-conventional shale gas in Vaca Muerta basin (El Pais, 2015) and the resulting reduction in fossil fuel prices. In addition, the use of the population as a driver for the spatial distribution of the emissions from some categories, such as RES, implies

another difficulty that is linked to the resolution used. Generally, international databases use country and subnational borders as proxy population data for the spatial disaggregation (Janssens-Maenhout et al., 2019), and this low-resolution population density, in comparison with that from locally developed inventories, may also be a source of spatial uncertainties. The latter is clearly noticeable not only in the local-global differences found by downscaling urban domains, but also in the spatial coverage of RES emissions in global inventories, where emissions are assigned even to large non-populated areas, such as the Amazon

rainforest or some desert areas of the region (Figure 4). Broader discussions on emissions from RES are given by Puliafito et al. (2021) and Álamos et al. (this issue) for Argentina and Chile, respectively.

For sources other than fuel combustion for energy purposes as many industrial processes, CAMS projections (based on CEDS trends) use population as driver (Hoesly et al., 2018). This approach may not be the most appropriate for many countries in the region, where changes in economic policies and even the occurrence of economic crises are frequent, affecting not

only consumption patterns but also the relocation of activities dependent on regional economies. Added to this, substantial differences have been observed in terms of the location of the IND sources in both inventories, probably attributable to the drivers used by the global databases for this purpose, with a very low presence of sources throughout the Argentine territory in CAMS, and a striking abundance of sources distributed over the central and northern region in Chile, which however do not reflect the heterogeneous share of emissions shown by the local distribution.

With the advancement of satellite tools, there are many aspects related to the spatial disaggregation of emissions that are improving over time, as the allocation of agricultural emissions. However, there are many aspects that the global inventory development methodologies have so far failed to replicate, whereas the incorporation of information generated from a local perspective has provided local inventories with greater precision for estimating emissions. In addition to the specific agricultural practices of the region, such as the predominant use of grazing for cattle farming or the use of large proportions of urea for the

fertilization of crops, during the last decades economic, natural and technological changes have occurred that have impacted on the sector's emissions and which are not reflected in global inventories (Castesana et al., 2018, 2020).





Lastly, and with the aim of contributing some aspects that may improve both global inventories and the dataset presented here, we list some information gaps identified when using the version CAMS-GLOB-ANT v4.1 as a base inventory for the region: (*i*) emissions from domestic and international civil aviation are not included; (*ii*) domestic and international waterborne

navigation are not reflected by downscaling the port city B. Blanca in Argentina; (*iii*) CAMS assigns zero FEF emissions for species other than NMVOCs in all the downscaled urban domains; (*iv*) in the whole region CAMS assigns zero emissions of NMVOCs from agricultural activities, both for those from animal excreta (manure managed-AGL and deposited in pasture-AGS) and those from crop production (AGS); and (*v*) CAMS assigns zero emissions of CO, $NO_x$ and $SO_2$ from SWD in Chile and Colombia even when the latter reports the occurrence of emissions in its local estimates. Item (*iii*) may be attributed to the

spatial distribution of FEF emissions in the global inventory, but given that this pattern is repeated in the six analysed domains and since in most of the urban domains considered there exist industrial facilities that may use venting and flaring to dispose their waste gases, the assumption of zero emissions for species other than NMVOCs seems to be more related to the omission of venting and flaring as emission sources under FEF (Granier et al., 2019). For its part, the item (*v*) brings to light the problems around waste management in the region, where waste is often disposed of in open dumpsites and, particularly regarding CO,

$NO_x$ and $SO_2$ emissions, uncontrolled open burning occurs, these practices are not reflected in local or global emission maps (UNEP, 2018). All these gaps in the base inventory were replicated in PAPILA dataset, except for the case of FEF emissions and NMVOCs emissions from agricultural activities in Argentina.

## 4  Data availability

Gridded maps with all inventories per species and year are available as netCDF4 (Network Common Data Format) files for

the regional domain (32°W–120°W, 34°N–58°S) at a resolution of 0.1° × 0.1° (PAPILA dataset) and can be accessed through the open access data repository http://dx.doi.org/10.17632/btf2mz4fhf.2, under a CC-BY 4 license (Castesana et al., 2021). The dataset includes all necessary attributes to be easily used in air quality simulations, such as molecular weight, projection and units. In addition, temporal profiles applied in the inventory evaluation for the Metropolitan Area of Buenos Aires, Argentina, are available in Appendix A of this work.

## 550  5  Conclusions

This work presents the results of the first joint effort of South American countries to generate regional maps of emissions inventories. Although it includes the emissions estimates made for a limited number of countries, it shows the enormous potential for the improvement of air quality studies in the region, by including local data into global databases and building together an inventory of the reactive gases CO, $NO_x$, NMVOCs, $NH_3$ and $SO_2$.

The results obtained through modelling using the inventory presented here as input are promising. Although there is room for improvements that may be linked to both emissions and other processes, the CO and $NO_x$ results in the Metropolitan Area



of Buenos Aires were able to reasonably reproduce surface observations, improving those obtained from the use of global inventories.

This work highlights the strengths and weaknesses of not only global inventories but also local ones. Although the latter
improve the representativeness of the estimates, the groups that generate information on emissions in the region do not necessarily have the same objectives (some are mainly oriented to the generation of input information for models, others towards mitigation measures that respond to air pollution concerns of their region) which is reflected in the uneven levels of development of local inventories. In this sense, it is worth mentioning that although the resources in the region often limit their growth, the capacities of the groups are growing, which is partly reflected in the development of local databases published in this same
special issue (Puliafito et al., 2021; Álamos et al., this issue; Osses et al., this issue).

In addition to individual advances, we want to emphasize the role of the PAPILA project and the EMISA initiative, which promote collaboration between groups in the region, enhancing efforts aimed at the development of appropriate and consistent surface emission inventories. In this auspicious context, we trust that this work will be a starting point for the development of comprehensive emission inventories in South America enriched with local information. To this end, the first step will be to
join the efforts of other countries in this endeavor, encouraging those with inventory capabilities to broaden their focus beyond cities by building national emission maps. In addition to the aspects that can be improved both in local and global inventories, we found it relevant to point out the key role of transparency in the allocation of sources in the categories for the development of coherent databases.

*Author contributions.* Paula Castesana: Conceptualization, Formal analysis, Visualization, Writing - original draft, review & editing. Melisa
Diaz Resquin: Conceptualization, Formal analysis, Visualization, Writing - original draft, review & editing. Nicolas Huneeus: Formal analysis, Writing - original draft, review & editing. Enrique Puliafito: Formal analysis, Writing - review & editing. Sabine Darras: Data processing. Darío Gomez: Formal analysis, Writing - review & editing. Claire Granier: Conceptualization. Mauricio Osses Alvarado: Writing - review & editing. Nestor Rojas: Writing - review & editing. Laura Dawidowski: Conceptualization, Formal analysis, Writing - original draft, review & editing.

*Competing interests.* The authors declare that they have no known competing financial interests or personal relationships that could have
appeared to influence the work reported in this paper.

*Acknowledgements.* This work was conducted within the framework of the Prediction of Air Pollution in Latin America and the Caribbean (PAPILA) project. The research leading to these results has received funding from the European Union H2020 program PAPILA (GA 777544). This work was carried out in part with the aid of grants PICT-O 2016-4802 (Agencia Nacional de Promoción Científica y
Tecnológica, Argentina) and PICT 2016-3590 (Fondo para la Investigación Científica y Tecnológica). We greatly acknowledge ECCAD (https://eccad3.sedoo.fr/) for the archiving and distribution of the datasets CNEA-3iA-GEIA (from Argentina), CR2-MMA (from Chile) and



CAMS-GLOB-ANT v4.1 used in this work. The authors also wish to thank the Environmental Protection Agency of Buenos Aires (APRA) for sharing the air quality data for this study.

## Appendix A: Supporting Material

**Table A1.** Description of sectors and codes considered in PAPILA inventory and their equivalences with CAMS sectors.

| PAPILA description | PAPILA code | IPCC 1996 code | IPCC 1996 description | CAMS 4.1 code |
|---|---|---|---|---|
| Thermal power plants | ENE | 1A1 | Energy industries | ENE |
| Residential, commercial and other combustion | RES | 1A4 | Residential, commercial and other combustion | RES |
| Road transportation | TRO | 1A3b | Road transportation | TRO |
| Non-road transportation | TNR | 1A3c, 1A3e | Rail and other transportation | TNR |
| Fugitive emissions | FEF | 1B1a, 1B2a1, 1B2a2, 1B2a3, 1B2a4, 1B2c, 7A | Coal Mining, Exploration, Production, Transport, Refining/Storage, Venting and Flaring | FEF |
| Industries (fuel combustion + refinering + industrial processes + product use) | IND | 1A1b, 1A1c, 1A2, 1A5, 1B1b, 1B2a5, 1B2a6, 1B2b5, 2, 3 | Petroleum refining, Manufacture of solid fuels and other energy industries, Other fuel combustion activities, Fuel combustion from manufacturing industries and construction, Fugitive emissions from solid fuel transformation, Fugitive emissions from distribution of oil products and gas natural, Industrial processes, Solvent and other product use. | IND + SLV |
| Agricultural soils | AGS | 4D | Agricultural soils: Synthetic fertilisers, Manure in pasture, Crops | AGS |
| Agricultural livestock | AGL | 4B | Manure Management | AGL |
| International ships | SHP-INT | 1A3d1 | International navigation | SHP |
| Ships | SHP | 1A3d2 | Inland navigation | SHP |
| Waste | SWD | 6 | Waste | SWD |



**Table A2.** Summary of $NO_x$ and NMVOCs total agricultural emissions by sector and categories for years 2014-2016.

| | | $NO_x$ (Gg y$^{-1}$) | | | NMVOCs (Gg y$^{-1}$) | | |
| --- | --- | --- | --- | --- | --- | --- | --- |
| | | 2014 | 2015 | 2016 | 2014 | 2015 | 2016 |
| | Manure management | | | | | | |
| | Dairy cattle | 3.2E-03 | 3.2E-03 | 3.2E-03 | 0.62 | 0.62 | 0.63 |
| AGL | Non-dairy cattle | 0.29 | 0.29 | 0.30 | 1.32 | 1.32 | 1.32 |
| | Other livestock | 0.95 | 0.96 | 0.94 | 11.04 | 11.08 | 10.92 |
| | **Total from AGL** | 1.24 | 1.25 | 1.24 | 12.98 | 13.03 | 12.86 |
| | Manure in pasture | | | | | | |
| | Dairy cattle | 7.80 | 7.73 | 7.38 | 1.25 | 1.25 | 1.25 |
| | Non-dairy cattle | 58.01 | 58.50 | 58.87 | 13.01 | 13.01 | 13.01 |
| | Other livestock | 12.88 | 13.55 | 13.64 | 0.10 | 0.11 | 0.11 |
| AGS | **Total from Manure in pasture** | 78.70 | 79.78 | 79.90 | 14.36 | 14.37 | 14.37 |
| | Fertilizers | 31.54 | 24.12 | 36.54 | NO | NO | NO |
| | Crops | NO | NO | NO | 29.88 | 34.23 | 33.73 |
| | **Total from AGS** | 110.24 | 103.90 | 116.44 | 44.24 | 48.60 | 48.10 |

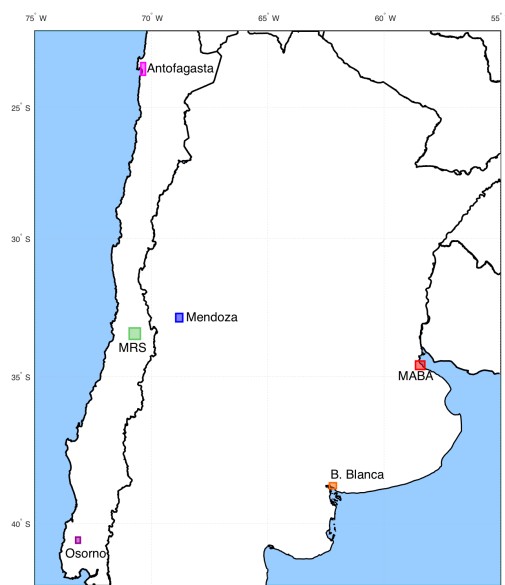

**Figure A1.** Location of the small domains analysed.



**Table A3.** Description of the selected domains.

| Urban domain | Population ($10^6$ inhab.) | Mean Temp. (daily max - daily min) [°C] | Characteristics | Coverage (km²) | Central coordinates |
|---|---|---|---|---|---|
| Antofagasta | 0.404 | 16.4° C (20.1 - 13.7) | PM and $SO_2$ Emissions from the copper extraction, smelting operations, heavy machinery operation, surface detonations and the transport of minerals in different degrees of fragmentation. 11k industries located in the area. | 680 (20x34) | 23.62º S; 70.3º W, 2260 m.a.s.l. |
| Metropolitan Region of Santiago de Chile (MRS) | 6.257 | 14.4° C (22.5 - 8.3) | It is a relatively large city located between two mountain ranges, the Andes on the east and the Coastal Range on the west. Pollution levels are very high in winter because of low wind speeds and strong temperature inversions (Gramsch et al., 2014). | 990 (30x33) | 33.74º S, 70.49º W, 567 m.a.s.l. |
| Osorno | 0.161 | 10.5° C (16.5-5.4) | Economy based on agriculture and livestock. 5k industries settled in the area. Cold region, residential/urban area. | 378 (18x21) | 40.53ºS; 73.18ºW, 32 m.a.s.l. |
| Metropolitan Area of Buenos Aires (MABA) | 13.3 | 17.9° C (9.5-23.4) | It is a Metropolitan area that concentrates 32% of the population of the entire country. It has several industrial poles and thermoelectric plants. | 1258 (37x34) | 34.57º S; 58.44º W, 14 m.a.s.l. |
| Bahía Blanca | 0.291 | 15° C (8-21) | City located near a Petrochemical Pole and a Port Area. | 572 (26x22) | 38.70º S; 62,22º W, 99 m.a.s.l. |
| Mendoza | 1.1 | 17.7° C (11.4-25) | It is one of the main cities of the country. It has a main industrial pole linked to the wine industry. | 924 (28x33) | 32.9º S; 68.7º W, 680 m.a.s.l. |




**Figure A2.** (left) Location of WRF-Chem Domains. Blue dots are the sites location. (right) Hourly, weekly and monthly variations used for the temporal disaggregation of emissions within this work.



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
