# Peer review of "PAPILA dataset: a regional emission inventory of reactive gases for South America based on the combination of local and global information"

_Earth System Science Data, 2021_

## Referee Comment (RC1)

Review of the manuscript "PAPILA dataset: a regional emission inventory of reactive gases for South America based on the combination of local and global information" by Castesana et al. ESSDD, 2021.

The paper describes the development of a comprehensive inventory of anthropogenic emissions of different gases for South America called PAPILA, taking as baseline the global database CAMS-GLOB-ANT v4.1 and enriching it with local information available for Argentina, Chile and Colombia, for the period 2014-2016. Differences at local and regional scales are analyzed and discussed for various geographical areas and emission sectors/categories. The work also provides a flowchart of a general methodology so that any relevant new or updated information can be easily added to the dataset in a standardized and consistent way. In addition, the paper compares the performance of the PAPILA and CAMS-GLOB-ANT v4.1 inventories by means of Air Quality simulations performed with WRF-chem model. They evaluate model results against in situ observations for Buenos Aires (Argentina) in summer and winter 2015, where PAPILA-based simulations showed slight improvements, mainly for the winter period. The authors have done a thorough and careful job in merging different information that has not been previously reported for South America, which is presented as a starting point of an international collaboration that represents a breakthrough for this community. The annual database provided is complete for the years 2014-2016, is accessible to download, and is organized in a user-friendly format. Based on this, I believe the paper could be published in ESSD after the following issues (mostly linguistic, but also technical and regarding conclusions) are revised.

Main Comments:

1) Language editing needs further investment. I am aware that this is mainly because English is not the native language of the authors, but in many places the writing style complicates comprehension and compromises the quality of the document. In addition, mostly within the Methods section, it is evident that different authors have contributed individually, and the text (and equations used) would benefit of using unified style. See specific comments below.

2) I understand that the local information from Colombia used in this study considers only a subset of species and categories in comparison with those from Argentina and Chile (Fig. 1). However, it would still be interesting to see a high-resolution comparison within an urban/industrial domain centered in Colombia into Table 2, to at least evaluate the impact

of using the proposed methodology which is similar but not identical to the one applied for Chile. Including a simplified description of the similarities and differences between Eq. 2 and Eq. 3 would also be useful.

3) Even though the results discussion is mostly focused on comparing the different contribution from the individual sectors for each species (Section3.1), I found a bit disproportioned the number of main Figures + Tables (2+1) comparing PAPILA and CAMS emissions in contrast to the Figures + Tables (also 2+1) focused on WRF-Chem air quality results. Note that the main focus of the paper is the development of the regional PAPILA inventory, and not a regional Air Quality study. Indeed, Section 3.1 compares emissions results within many different urban/industrial local domains within Argentina and Chile (Table 2), but the WRF-Chem analysis is centered only over Buenos Aires. Thus, the WRF-Chem simulation in MABA should be explicitly presented as a single case study analysis, and explicitly mention that the improvements with respect of considering CAMS emissions might not be applicable to the other selected urban/industrial areas within Argentina, Chile and Colombia (which otherwise would require a much larger description and evaluation of the WRF-Chem setup).

4) Section 5 (Conclusions) is a bit vague, it includes several adjectives that are not commonly used in scientific works (enormous, promising, auspicious, etc.) and focus on highlighting the cooperative effort of a South American community to develop emissions inventories and air quality research. However, the authors do not provide neither arguments supporting the main differences, strengths and/or weaknesses among PAPILA and CAMS, nor suggestions for future improvements of the PAPILA dataset. In other words, I would also expect to summarize in the conclusions the main methodological approaches used in the development of the PAPILA inventory, as well as the most important results of considering an improved inventory with local and high-resolution data. The current conclusion section seems to belong to another paper, or to the main benefits of a research proposal.

5) Finally, I would like to make a personal suggestion (not mandatory but that might increase the usefulness of the PAPILA inventory as input for air quality models): Could you include aerosol information (i.e., PM10 emissions), either from local or global inventories, into the PAPILA dataset? Following the methodology described in this paper, I believe it should be possible. Indeed, you have done so to perform the WRF-Chem simulations in this study based on EDGAR and CAMS. Having said this, I understand this might not be possible at

present time (due to data availability or even due to time dedicated to this project) and might be included into the 2nd version of the PAPILA dataset. In addition, a comparison between the PAPILA inventory and satellite information would also be interesting for future (or the current) work.

Minor comments:

1. L11: I found no need to explicitly mentioning the DOI for the PAPILA dataset on the Abstract. Also, I suggest using evaluation instead of assessment when the comparison between PAPILA and CAMS is mentioned.

2. L39: Please check if a more updated reference than 2002 is available on this topic. Country restrictions may have changed in the last 18 years.

3. L102: Figure 2 is quoted in the text before Figure 1 (L130). I suggest avoiding pointing at a Figure within the introduction.

4. L161: I do not understand why the trends for Chile and Colombia are specified within the Argentina section.

5. L202: Please add a reference for this statement

6. Table1: Consider restructuring this table, I found it quite difficult to read.

7. L211: Make reference to the results of section 3.1 where this topic is discussed

8. L263: Specify the version of WRF-Chem used as well as whether you used any spin up time in the simulation.

9. L289: Since PM data is mentioned as existing, why is it not further used for the model validation?

10. L288: Is there any reference that used this iterative method or any previous study that contains a description of a similar methodology? Was the iterative method applied only to WRF-Chem simulations based on PAPILA or also when CAMS was used? In case you applied to both, did you get similar results?

11. Figure 3: The description of the abbreviations is found much further back in the text (pg 4 and 6), it might be useful reinserting it in the caption of this figure together with a clarification of the sectors included in "Others" for easier interpretation of the figure.

12. L345: Please refer to the database used. If the SHIP-INTs for B. Blanca do not appear in CAMS, from which global estimate were they extracted? It is not clear.

13. L437: Clarify the meaning of the abbreviation "S-emitting", not mentioned above.

14. Figure 5: I would recommend placing a dot with the average value of each time series aside from the median that is already in the plot.

15. L464: Add the pollutants to which the hourly concentrations correspond to, for more clarity in the sentence.

16. L572: The document mentions several times "transparency", but never really specifies what it actually entails.

17. The "regions" variable of the netcdf files is not described in the metadata nor in the general description of the readme.txt, intuitively we would say that they are time zones but it is not so, please clarify it.

Language editing comments:

19. L19: Change "relative to" by "in comparison to" or "in relation to"

20. L42: the word wood is repeated twice.

21. L79: PAPILA acronym is used in the introduction before it is defined.

22. L90: What is LAC?

23. L156: The sentence is confusing, please rephrase it.

24. L170: Equation indexes and styles for the Argentine emissions are not the same as the one used for Chile and Colombia. Please unify.

25. L 189: The sentence is too long, break it down into shorter, more specific segments.

26. L204: Check punctuation marks, the sentence is too long.

27. L213: Please rephrase

28. L220: Throughout the text, sometimes the term sectors is used and in some other you refer to categories. Please unify and, in case there is any difference among them, it should be explicitly mentioned.

29. L227: This sentence could be written in a more concise way.

30. L235: This sentence could be written in a more concise way.

31. L237: The word "such" is redundant.

32. L257: MABA is the City of Buenos Aires? Please define.

33. L427: The whole paragraph contains very long and repetitive sentences.

34. L460: I recommend the use of a more technical language to present the results e.g.: "errors in PAPILA results decreased in winter".

35. L483: Please rephrase.

36. L496: change "often resorted to" by "often used"

37. 559: The sentence is complex, please rephrase

38. Figure A2: change sites by "monitoring sites"

---

## Referee Comment (RC2)

Review of essd-2021-208 PAPILA dataset: a regional emission inventory of reactive gases for South America based on the combination of local and global information by Paula Castesana et al.

The paper describes a dataset intended on providing air quality and climate modellers with a complete dataset for South America (SA) by creating a so-called mosaic inventory. This implies using a complete but rather generic global dataset with less granularity than local or national data as a starting point and merging it with more detailed national scale inventories. The result of such a mosaic inventory still provides a complete dataset but with higher granularity and includes more local knowledge thereby providing modellers with a better starting point for their model exercises which can lead to more accurate (scientific) results and analysis. The merged dataset needs to be carefully evaluated and discrepancies explained and documented as a risk is that "apples and oranges" are treated in the same way and the end result can also be confusion. The advantage of the approach as also stressed by the authors is that the dataset can improve over time incorporating more data as they become available and the current PAPILA dataset is not intended nor expected to be the final dataset but a well-documented starting point that can improve over the years.

As said such datasets can be an asset for modellers and support / improve regional air quality analysis. The topic is fitting for ESSD and the paper could be published after several clarifications and improvements are made.   .

My main concerns are on the discussion of the results and clarity of the associated messages.

- The idea of the mosaic approach is not new and has been successfully applied in the framework of theTask Force on Hemispheric Transport of Air Pollution (TF HTAP). In the introduction it would be good to refer to the HTAP v2.2 dataset as an example (Maenhout et al., 2015) including the fact that this dataset has been widely used also outside of HTAP. This could be added in the paragraph L68-73
- The discussion on the impact of the incorporation of the national/local data is unintentionally misleading. For example in L339 "resulting in a difference for SA of only -5%". Strictly speaking this is true BUT you have only replaced 3 countries of total SA and not even for all pollutants (Figure 1). When looking at Table 2 it can be deduced that these 3 countries are only responsible for 20-25% of the total SA emissions (in the case of CO). Thus over 75% of the emissions  are unchanged and hence the impact on total SA is very limited. It should be explained in this way and then  followed by what the impact is on the three countries that were adjusted. (For CO this is ~-20% which is substantially more than the -5% for all SA, and in fact is still a bit misleading because the countries compensate for each other but this is properly stated in L338. )  This applies to all species (e.g. L365 NOx – same story, this 3% is caused by the bulk of the emission being unchanged because a.o. Brazil and Venezuela are kept constant)
- Modelling is done for MABA (Buenos Aires) only. It should be more clearly stated in the conclusions that these results do not give any insight in the performance for e.g. Colombia, Chile etc. SA is such a big area that a better performance over MABA cannot be taken as sign that also in other countries local data result in better AQ modelling results. I agree it is what we may expect / hope for but it is no prove whatsoever. As an outlook it could be mentioned that an evaluation of the PAPILA dataset over other regions where local data are integrated is not only highly recommended but truly needed to prove the added value of the approach. (see your conclusion section)

- L116 – the shipping might need a bit more discussion. Inland navigation is clear but the domestic coastal (and what do you mean with deep-sea ? ) cannot be so easily separated from SHP-INT – how do you do this? Is that not leading to double counting? SHP-INT will be based on AIS signals but no separation is made between flag states, so how will e.g. Chilean coastal shipping be distinguished from SHP-INT? .
- L398. The main point is that CAMS treats countries uniform w/o correcting for the climatic zones in the country. However, SA countries are large and the climatic / temperature zones vary widely from tropics to antartica. In the discussion this could be mentioned and suggested as an improvement for global inventories as the temperature zones are well known and available on grids. Theis could be used to redistribute within a country.
- L504 – here and throughout the MS – where you use "driver" you mean "proxy". (Economic growth is a driver for growing emissions; population density is a proxy to distribute such emissions). Please check the whole section 3.3. and further for this terminology.
- The discussion in L505-511 is unclear, please rewrite and check for using the right words. (it can be a low-resolution map but not low-resolution density)
- L512 as should be or? ; population as a driver should be "population density as a proxy"
- L562 remove "which is reflected in the uneven levels of development of local inventories" as such it does not add information.
- Final paragraph – I think you can add that especially Brazil and 1 or 2 others should be added because the 3 countries added now are responsible for less than 25% of the emissions. How would this change if Brazil is addd?
- L571 sentence "in addition etc. can be removed, the sentence before is a stronger ending with a clear message.

Table A2 – is that really a summary of NOx ? or do you mean NH3? I'm surprised to see NOx emissions come from dairy cattle.

**Minor issues**

L5 place "derived" after "data"

L13 "obtained" should be "found"

L15 ….lower **levels**….

L20 replace: PAPILA-based modelling results had a lower bias for CO and NOx concentrations in winter while CAMS-based results for the same period tended to deliver an underestimation of these concentrations.

L28 From the energy standpoint  ->     Regarding energy use,

L39 it's a bit awkward to use a reference of 20 years ago here. Much would / may have hanged since then. I think this is also discussed in Huneeus et al. 2020; might be a added here?

L43 change larger to increasing

L77 change bibliography to literature

L91 choose either LAC or SA and use only one. Please check full MS and SI for this

L98 2 times have

L99 Explain a bit better why you need complete country inventories (because this is the common / shared administrative entity that can be exchanged with the global inventory)

L140 – yes that is a solution but it is not really necessary, the combination of coordinates AND a country code could also generate unique values and the cell can than be shared by 2 or even 3 countries. The difficulty is probably that the global inventory does not have these country codes

Figure 1 – I would call that a Table and not a figure. Replace SA on the bottom with "Rest of SA" or "Other SA"

L156 explain how you distributed the rest as area sources? With what proxy?

L179 taken to  -> converted to

L183 Such => This

L191 considered => assumed

L215 check word missing (we to report?)

L228 applied => apply

L378 minority => minor

Table 2. it would  be good to indicate that the first 3 subregions are in Argentina and the last  3 in Chile. This is well explained in the text but for international readers it might still be difficult to remember this. Moreover it should be considered (but only a suggestion) if it is not better to e.g. in L250 where MABA is introduced to simply state that in this study Buenos Aires includes the entire Metropolitan area of BA. In the paper you can then discuss simply  Buenos Aires and do not have to use this MABA which for many people will be something they are not familiar with. Same applies to MRS. Everybody knows Santiago is the capital of Chile, MRS will not be known to many. Also in the Table 2 acronyms like MABA and MRS are not very helpful. You could add a table footnote that the entire metropolitan region is included and use the city names in the table.

Table 2 – I suggest to only show a digit if the value is smaller than 100.  E.g. 31225.2 is not very helpful. PAPILA NOx column has 2 times 16.8 – pls check if correct

Fig 3 . Pls consider if it is not better to change the order and show Argentina – followed by its 3 sub regions  (same for Chile). This may help to see patterns. For example when looking at SO2 the pattern for ARG and sub region is exactly the same (so it's a national consistent difference) for Chile it is different with a redistribution between Antofagasta and MRS.

L415. How relevant are these emissions?  NH3 is dominated by AGRI; 80% sounds big but if 80% of almost nothing is still nothing. Is it worth the attention?

L430 non => not

L435 see earlier remark on comparison for the entire region (which is not informative as bulk of the emissions are taken from CAMS and thus constant)

Table 3 – explain acronyms in table footnote (NMB, NMGE, FAC2)

L457 null => no

L460 goodness of the => better

L467 high => height

Fig 6 change top legend : remove "simul" (it is not a known word), make it bigger and have e.g. CAMS simulation; PAPILA simulation

L477 goodness of => quality

L491 than => as

L495 what is intended with hydraulic availability??? Change and explain better, different word?

L497 the incorporation of diesel-based motor generators => the use of diesel-fuelled generators

L504-505 Why does this imply a difficulty?

L520 tools => monitoring

L520-527 – Please rewrite this paragraph and check with co-authors; it is rather cryptic. E.g. "information generated from a local perspective" can simply be "local information"

L529 Why are aviation emissions not included? Are they not important? Not provided by EDGAR or CEDS?

L534 item v) – Please explain this better and be a bit more informative on its relevance – are these emissions missing or possibly reported elsewhere? If it is CO, NOX, SO2 it must be from waste burning not landfills or waste water. Is this not present for all SA counties? Or only for Chile and Columbia? When you look at the local information where you have these emissions included, how important is it? What is the share for the country or city? If it is missing but only adds e.g. 0.1% to the national total is it relevant?

L541 replicated in => addressed in the

L541 remove "for the case of"

Section 4 – I did not check the dataset and the sums with the tables. Please ask one of your 10 co-authors who was not involved in uploading the data to download and do a check. I assume it will be good but advise to do this check.

L552 made for a limited number of countries – change to "only for Argentina, Chile and Colombia" (be as specific as possible) and add that this is a living dataset and that in the future other countries can be added.

**Refs**

Janssens-Maenhout, G., Crippa, M., Guizzardi, D., Dentener, F., Muntean, M., Pouliot, G., Keating, T., Zhang, Q., Kurokawa, J., Wankmüller, R., Denier van der Gon, H., Kuenen, J. J. P., Klimont, Z., Frost, G., Darras, S., Koffi, B., and Li, M.: HTAP_v2.2: a mosaic of regional and global emission grid maps for 2008 and 2010 to study hemispheric transport of air pollution, Atmos. Chem. Phys., 15, 11411–11432, https://doi.org/10.5194/acp-15-11411-2015, 2015.

---

## Author Comment (AC1)

| | |
|---|---|
| The paper describes the development of a comprehensive inventory of anthropogenic emissions of different gases for South America called PAPILA, taking as baseline the global database CAMS-GLOB-ANT v4.1 and enriching it with local information available for Argentina, Chile and Colombia, for the period 2014-2016. Differences at local and regional scales are analyzed and discussed for various geographical areas and emission sectors/categories. The work also provides a flowchart of a general methodology so that any relevant new or updated information can be easily added to the dataset in a standardized and consistent way. In addition, the paper compares the performance of the PAPILA and CAMS-GLOB-ANT v4.1 inventories by means of Air Quality simulations performed with WRF-chem model. They evaluate model results against in situ observations for Buenos Aires (Argentina) in summer and winter 2015, where PAPILA-based simulations showed slight improvements, mainly for the winter period. The authors have done a thorough and careful job in merging different information that has not been previously reported for South America, which is presented as a starting point of an international collaboration that represents a breakthrough for this community. The annual database provided is complete for the years 2014-2016, is accessible to download, and is organized in a user-friendly format. Based on this, I believe the paper could be published in ESSD after the following issues (mostly linguistic, but also technical and regarding conclusions) are revised. | Dear Rafael, We welcome your comments and suggestions, which have helped us improve our manuscript. We trust that we have responded satisfactorily to all comments in this document. |
| **Main Comments:** Q1 Language editing needs further investment. I am aware that this is mainly because English is not the native language of the authors, but in many places the writing style complicates comprehension and compromises the quality of the document. In addition, mostly within the Methods section, it is evident that different authors have contributed individually, and the text (and equations used) would benefit of using unified style. See specific comments below. | We have improved all the language and style issues noted by the reviewer. In addition, an English speaker will unify the language of the revised version of our manuscript. |
| Q2 I understand that the local information from Colombia used in this study considers only a subset of species and categories in comparison with those from Argentina and Chile (Fig. 1). However, it would still be interesting to see a | The analysis of the differences between the emissions reported by local and global inventories for small domains was done in this work only for those local inventories that have implemented their own methodologies for the spatial distribution of emissions. In some cases, these methodologies involve the use of activity data already disaggregated, and in others implementing the |

| | |
|---|---|
| high-resolution comparison within an urban/industrial domain centered in Colombia into Table 2, to at least evaluate the impact of using the proposed methodology which is similar but not identical to the one applied for Chile. Including a simplified description of the similarities and differences between Eq. 2 and Eq. 3 would also be useful. | use of specific proxies, not necessarily the same as those used by global inventories. The PAPILA/CAMS comparative analysis in urban domains evaluates and compares the applied spatial distribution criteria. This is not the case in Colombia, for which the national inventory lacks an implemented spatial disaggregation methodology, and therefore only a comparison was made at the national level finding no advantages to do it for smaller domains. However, from this observation we have noticed that the reason for our decision has not been sufficiently clear in our original manuscript, and we have decided to better clarify it in the last paragraph of section 2.3, replacing the original phrase "(*iii*) urban domains from those countries with local information on the spatial disaggregation of emissions" by "(*iii*) urban domains from those countries that have implemented their own methodologies for the spatial distribution of emissions". In addition, in order to clarify the description of the Eq. 3, we replaced the phrase "using the spatial distribution of sources of the base inventory" in line 223 of the original manuscript was by "using the spatial distribution of sources of the CAMS inventory", while the Eq. 2 is based on the spatial distribution of the Chilean inventory (described in lines 193-194 of the original manuscript). |
| **Q3** Even though the results discussion is mostly focused on comparing the different contribution from the individual sectors for each species Section3.1), I found a bit disproportioned the number of main Figures + Tables (2+1) comparing PAPILA and CAMS emissions in contrast to the Figures + Tables (also 2+1) focused on WRF-Chem air quality results. Note that the main focus of the paper is the development of the regional PAPILA inventory, and not a regional Air Quality study. Indeed, Section 3.1 compares emissions results within many different urban/industrial local domains within Argentina and Chile (Table 2), but the WRF-Chem analysis is centered only over Buenos Aires. Thus, the WRF-Chem simulation in Buenos Aires should be explicitly presented as a single case study analysis, and explicitly mention that the improvements with respect of considering CAMS emissions might not be applicable to the other selected urban/industrial areas within Argentina, Chile and Colombia (which otherwise would require a much larger description and evaluation of the WRF-Chem setup). | Thank you for this comment. As the reviewer mentioned, the intention of the simulations made with WRF-Chem in Buenos Aires was to present a single case study analysis. With this in mind, we have tried to maintain a balance in the manuscript between the main objective of the work and this evaluation exercise, using 2 of the 11 pages of the Results and Discussion section in the original manuscript. We agree with the reviewer that it is necessary to highlight in the text that a broader evaluation is still needed, evaluating the PAPILA dataset in the other regions where local data were integrated in global datasets. To address that, we will include some comments in the general modifications that will be made in the conclusions (see Q4), and the following changes in the other sections: Lines 255-258: the text "The performance of the PAPILA dataset in comparison with that of CAMS as input data to air quality models was assessed using the Weather Research and Forecasting Chemistry (WRF-Chem v4.1.2) regional model. The site chosen for this case study was Buenos Aires, a megacity strongly influenced…", was replaced by: "The performance of the PAPILA dataset in comparison with CAMS can be assessed using both inventories as input data of a regional model, implemented in the whole domain where local data has been integrated into the global dataset. This vast region, that includes the tropical Andes in Colombia, the dry Andes in Southern Chile and the Argentinean plateau towards the Atlantic coast, is characterized by a diverse topographic features and vegetation patterns. In order to capture the differences in boundary layer process and surface energy budget in the whole area, a high-resolution model is needed, setup in each area where the main PAPILA/CAMS datasets changes have been made. As a first step of this verification exercise, we present here a study focused on Buenos Aires using the Weather Research and Forecasting Chemistry regional model version 4.1.2 (WRF-Chem v4.1.2). This megacity is strongly influenced…" In addition, the title of the subsection 2.4 was modified as follows: "WRF-Chem Simulations: case study in Buenos Aires", and the title of the subsection 3.2 was replaced by "Case study: model evaluation and results". |
| **Q4** | |

| | |
|---|---|
| Section 5 (Conclusions) is a bit vague, it includes several adjectives that are not commonly used in scientific works (enormous, promising, auspicious, etc.) and focus on highlighting the cooperative effort of a South American community to develop emissions inventories and air quality research. However, the authors do not provide neither arguments supporting the main differences, strengths and/or weaknesses among PAPILA and CAMS, nor suggestions for future improvements of the PAPILA dataset. In other words, I would also expect to summarize in the conclusions the main methodological approaches used in the development of the PAPILA inventory, as well as the most important results of considering an improved inventory with local and high-resolution data. The current conclusion section seems to belong to another paper, or to the main benefits of a research proposal. | The conclusions will be modified and expanded as suggested by the reviewer. |
| **Q5**

Finally, I would like to make a personal suggestion (not mandatory but that might increase the usefulness of the PAPILA inventory as input for air quality models): Could you include aerosol information (i.e., PM10 emissions), either from local or global inventories, into the PAPILA dataset? Following the methodology described in this paper, I believe it should be possible. Indeed, you have done so to perform the WRF-Chem simulations in this study based on EDGAR and CAMS. Having said this, I understand this might not be possible at present time (due to data availability or even due to time dedicated to this project) and might be included into the 2nd version of the PAPILA dataset. In addition, a comparison between the PAPILA inventory and satellite information would also be interesting for future (or the current) work. | We agree to expand the presented inventory of reactive gases by incorporating particles, and the idea is to do so in a future version. In relation to the comparison with satellite information, the idea of this project is to improve global inventories with local data, prior to the remote sensing and surface data assimilation exercise. |
| **Minor comments:**

**Q6**

L11: I found no need to explicitly mentioning the DOI for the PAPILA dataset on the Abstract. Also, I suggest using evaluation instead of assessment when the comparison between PAPILA and CAMS is mentioned. | We included the DOI following the instructions specified in https://www.earth-system-science-data.net/submission.html#assets. |
| **Q7**

L39: Please check if a more updated reference than 2002 is available on this topic. Country restrictions may have changed in the last 18 years. | Modified as suggested: Huneeus, N., Denier van der Gon, H., Castesana, P., Menares, C., Granier, C., Granier, L., Alonso, M., de Fatima Andrade, M., Dawidowski, L., Gallardo, L., Gomez, D., Klimont, Z., Janssens-Maenhout, G., Osses, M., Puliafito, S. E., Rojas, N., Sánchez-Ccoyllo, O., Tolvett, S., and Ynoue, R. Y.: Evaluation of anthropogenic air pollutant emission inventories |

| | for South America at national and city scale, Atmos. Env., 235, 117 606, https://doi.org/https://doi.org/10.1016/j.atmosenv.2020.117606, 2020a. |
|---|---|
| **Q8**

L102: Figure 2 is quoted in the text before Figure 1 (L130). I suggest avoiding pointing at a Figure within the introduction. | Thanks for this observation. To avoid this inconsistency, we have decided to delete the quote of the Figure 2 on line 102 of the original manuscript. |
| **Q9**

L161: I do not understand why the trends for Chile and Colombia are specified within the Argentina section. | We appreciate this observation and apologize for the mistake which has already been corrected in the manuscript. |
| **Q10**

L202: Please add a reference for this statement | We added the reference Huneeus, N.et al (2020b): Informe a las Naciones. El aire que respiramos: pasado, presente, futuro. Contaminación atmosférica por MP2,5 en el centro y sur de Chile, available in www.cr2.cl/contaminacion/. |
| **Q11**

Table1: Consider restructuring this table, I found it quite difficult to read. | The Table was restructured. |
| **Q12**

L211: Make reference to the results of section 3.1 where this topic is discussed | The reference was added. |
| **Q13**

L263: Specify the version of WRF-Chem used as well as whether you used any spin up time in the simulation. | In the original manuscript, the WRF-Chem version is specified in line 256 as "Weather Research and Forecasting Chemistry (WRF-Chem v4.1.2) regional model". From this suggestion, we have modified the phrase as follows: "Forecasting Chemistry regional model version 4.1.2 (WRF-Chem v4.1.2)". The spin up period was added at the end of the first paragraph of the subsection 2.4.1: "All the simulations conducted in this study were performed using a spin up time of two weeks". |
| **Q14**

L289: Since PM data is mentioned as existing, why is it not further used for the model validation? | As this first version of the PAPILA dataset does not include PM emissions, we present in the article only the pollutants included in the dataset for which there was air quality data in the two monitoring stations in the city of Buenos Aires, which are CO and $NO_x$. Aerosols were added to the model to use the chemical scheme that was already tested for the region during the previous simulations, mentioned in item 10, used to adapt the diurnal cycle. |
| **Q15**

L288: Is there any reference that used this iterative method or any previous study that contains a description of a similar methodology? Was the iterative method applied only to WRF-Chem simulations based on PAPILA or also when CAMS was used? In case you applied to both, did you get similar results? | Thank you for this observation. From this observation we found that indeed the term "iterative" is not adequate to describe what has been done. We will replace the sentence in lines 287-290 of the original manuscript with the following: "The diurnal cycles were adapted from those reported by Wang et al. (2010), focusing on reproducing Buenos Aires's traffic patterns observed in the two monitoring stations: Parque Centenario and Córdoba". The process of adapting the diurnal cycles is described in greater detail in another article that has not yet been published. The process aimed to adapt the Wang et al. (2010) cycles so that the maximum and minimum traffic levels match with those of |

| | Buenos Aires, using Puliafito et al. (2017) and EDGARv4.1 emission inventories. Similar cycles were obtained by using both inventories. |
|---|---|
| **Q16**

Figure 3: The description of the abbreviations is found much further back in the text (pg 4 and 6), it might be useful reinserting it in the caption of this figure together with a clarification of the sectors included in "Others" for easier interpretation of the figure. | The caption was modified by adding "ENE + IND: energy and industries; RES: residential and commercial combustion; TRO: road transportation; Others: non-road transportation, fugitive emissions, agricultural soils, agriculture livestock, navigation and waste". |
| **Q17**

L345: Please refer to the database used. If the SHIP-INTs for B. Blanca do not appear in CAMS, from which global estimate were they extracted? It is not clear. | The dataset used from the coastline of each country outwards is the CAMS. In line 345 we mean that we have not detected activity from the Bahía Blanca port to the offshore in the global inventory (the local inventory does not include emissions outside the coastline). However, from the review process we understood that we have not been precise when describing what is related to navigation, For that reason, this paragraph will be modified to better explain what was done also taking into account the modifications to be made from Q4 of the Reviewer #2. |
| **Q18**

L437: Clarify the meaning of the abbreviation "S-emitting", not mentioned above. | "S-emitting" was replaced by "sulfur emitting industries". |
| **Q19**

Figure 5: I would recommend placing a dot with the average value of each time series aside from the median that is already in the plot. | Following your suggestion, we placed a dot with the average values, and we changed the caption to explain this. |
| **Q20**

L464: Add the pollutants to which the hourly concentrations correspond to, for more clarity in the sentence. | Modified as suggested. |
| **Q21**

L572: The document mentions several times "transparency", but never really specifies what it actually entails. | Transparency is one of the indicators of inventory quality defined by the IPCC, and for this reason it is a well-known term in inventory development. In section 1.4: Inventory quality of Volume 1: General Guidance and Reporting of the IPCC 2006 guidelines, it is defined:

Transparency: There is sufficient and clear documentation such that individuals or groups other than the inventory compilers can understand how the inventory was compiled and can assure themselves it meets the good practice requirements for national greenhouse gas emissions inventories.

CHAPTER 1 INTRODUCTION TO THE 2006 GUIDELINES |
| **Q22**

The "regions" variable of the netcdf files is not described in the metadata nor in the general description of the readme.txt, intuitively we would say that they are time zones but it is not so, please clarify it. | It is true that the variable "regions" is not described in the dataset. We have decided to remove it from the dataset since it does not add information to it. The final product resulting from the full review process will be published as http://dx.doi.org/10.17632/btf2mz4fhf.3 instead of http://dx.doi.org/10.17632/btf2mz4fhf.2. |

| Language editing comments: | |
|---|---|
| Q23
L19: Change "relative to" by "in comparison to" or "in relation to" | Modified as suggested. |
| Q24
L42: the word wood is repeated twice. | Modified as suggested. |
| Q25
L79: PAPILA acronym is used in the introduction before it is defined. | In the line 79 of the original manuscript the term PAPILA is not an acronym but acts as the name of the dataset ("The dataset presented in this work, hereinafter called PAPILA…"). The acronym is clarified in the first line of the abstract and in line 88 of the original manuscript (Introduction section). |
| Q26
L90: What is LAC? | The acronym LAC was removed as it is not mentioned in the rest of the manuscript. Instead we have written "Latin America and the Caribbean". |
| Q27
L156: The sentence is confusing, please rephrase it. | The paragraph in the original manuscript: "The GEAA inventory has been updated for this work including emissions from IND, which were not covered in the published (Puliafito et al., 2017). With these changes for manufacturing industries, the dataset considers fuel consumption by fuels, petroleum refining and emissions from the production process itself for the main industries, spatially distributed with the location of the main industries and distributing the rest as area sources in the whole territory. In all these categories the combustion of fossil and biomass fuels was considered.", was replaced by: "The GEAA inventory has been updated for this work including emissions from IND, which were not covered in the published (Puliafito et al., 2017). These emissions include (*i*) those from fuel consumption and from production process itself for the main industries, disaggregated by fuel and spatially distributed with the precise location of each facility, and (*ii*) those from fuel consumption of small industries, whose consumption is known by activity and by district, and whose spatial disaggregation of emissions was carried out using the population density of each district as a proxy". |
| Q28
L170: Equation indexes and styles for the Argentine emissions are not the same as the one used for Chile and Colombia. Please unify. | In a unified way throughout the manuscript we have used indexes *i* to refer to species, *j* for categories, and *k* for cell grid when applicable. The difference in styles between the equations of Argentina with respect to those of Chile and Colombia is the following: for Argentina, we are showing the expression applied to estimate the emissions of each species *i*, whereas the equations for Chile and Colombia show the spatial disaggregation methodology of emissions that have already been estimated. |
| Q29
L 189: The sentence is too long, break it down into shorter, more specific segments. | The original sentence "However, given that the local methodology for $SO_2$ emission estimates is based on sulfur content in fuels and in mass-flow balances in copper production processes, which constitute the main $SO_2$ emitter activity in Chile (Gonzalez Rojas, 2021), we have considered that the information on sulfur content that is handled locally is reliable, and included |

| | the spatially distributed emissions as estimated in Chile in our dataset." was broken down as follows: |
| --- | --- |
| | "For the particular case of SO$_2$, the local methodology for the emission estimates is based on sulfur content in fuels and in mass-flow balances in copper production processes, which constitute the main SO$_2$ emitter activity in Chile (Gonzalez Rojas, 2021). For this reason, and assuming that the information on sulfur content that is handled locally is reliable, we have included the spatially distributed emissions as estimated in Chile in our dataset." |
| **Q30** L204: Check punctuation marks, the sentence is too long. | A comma was added to make it easier to read. |
| **Q31** L213: Please rephrase | The paragraph "Emissions of CO and NO$_x$ from urban and non-urban road transportation were added under the TRO category. Given that the local inventory reports ENE and IND (including use of solvent) emissions together and that insufficient information foro spatial disaggregation was available, we to report ENE + IND under the IND sector for the case of Chile" was rephrased as "Local estimates of CO and NO$_x$ emissions from urban and non-urban road transportation were aggregated and reported in PAPILA dataset under the TRO category. Given that both the magnitudes and the spatial distribution of emissions from ENE and IND (including use of solvent) are reported in an aggregate way in the Chilean inventory, we decided to report them under the IND category". |
| **Q32** L220: Throughout the text, sometimes the term sectors is used and in some others you refer to categories. Please unify and, in case there is any difference among them, it should be explicitly mentioned. | We appreciate this observation and apologize for the mistake which has already been corrected in the manuscript, unifying the use of the term "categories". |
| **Q33** L227: This sentence could be written in a more concise way. | The original paragraph "Although in this context the country estimates CO, NO$_x$ and SO$_2$ emissions from solid waste, wastewater and waste incineration, SWD emissions were taken from CAMS. The reason for this decision was that although the magnitude of the emissions was available, there was no information on their spatial distribution and it was not possible to apply the methodology described above, since CAMS considers zero SWD emissions for these species in Colombia" was replaced by "Although in this context the country reports CO, NO$_x$ and SO$_2$ emissions from SWD, CAMS reports them as zero. The latter precluded the spatial assignment of the locally estimated emissions, and for this reason it was decided to take the SWD category from the CAMS". |
| **Q34** L235: This sentence could be written in a more concise way. | The original sentence "However, we only need to compare two inventories and are also interested in observing the differences in terms of magnitude, we therefore propose a comparison of normalized emissions by category and urban domain normalizing them with respect to those from the CAMS dataset, such as shown in Eq. 4" was replaced by "Since in our work we are interested in comparing only two inventories without losing sight of the differences in |

| | terms of magnitude, we have adapted this approach by comparing normalized emissions by category and urban domain, normalizing them with respect to those from the CAMS dataset as shown in Eq. 4". |
|---|---|
| Q35

L237: The word "such" is redundant. | The word "such" was removed. |
| Q36

L257: MABA is the City of Buenos Aires? Please define. | MABA corresponds to the Metropolitan Area of Buenos Aires, and the acronym is defined in line 250 of the original manuscript. However, we note that this acronym is not very reader-friendly, and at the suggestion of another reviewer we have replaced it "Buenos Aires", clarifying that Buenos Aires will refer to the big area of the MABA in our article. |
| Q37

L427: The whole paragraph contains very long and repetitive sentences. | As suggested, and also in accordance with what was indicated by the reviewer # 2, we will modify the text in the revised version of our manuscript. |
| Q38

L460: I recommend the use of a more technical language to present the results e.g.: "errors in PAPILA results decreased in winter". | The sentence "Thus, the goodness of the PAPILA-based results exhibited for winter were not that apparent for summer" in the original manuscript was replaced by "Thus, the results for the summer simulations were not as conclusive as for the winter simulations". |
| Q39

L483: Please rephrase. | The original paragraph "based on the estimates of the EDGARv4.3.2 of the year 2012. This is different from what has been done in this work, since for the three countries 2014 was taken as the base year, and while the same methodology than that used by CAMS was applied to extrapolate to 2015 and 2016 for Chile and some categories for Argentina, locally estimated trends were applied for Colombia" was modified as follows: "In contrast, in our work there were three different situations: (*1*) for Colombia, locally estimated trends were applied based on 2014 local emission estimates, (*2*) for some categories in Argentina we have applied annual estimates for the entire period, and (*3*) for the rest of the categories in Argentina and for the Chilean inventory the same methodology than that used by CAMS was applied, but based on local estimates for 2014". |
| Q40

L496: change "often resorted to" by "often used" | Modified as suggested. |
| Q41

559: The sentence is complex, please rephrase | In the revised version of our manuscript we will restructure the conclusions as suggested in Q4, and we will consider this observation. |
| Q42

Figure A2: change sites by "monitoring sites" | Modified as suggested. |

---

## Author Comment (AC3)

| | Dear Hugo, |
|---|---|
| The paper describes a dataset intended on providing air quality and climate modellers with a complete dataset for South America (SA) by creating a so-called mosaic inventory. This implies using a complete but rather generic global dataset with less granularity than local or national data as a starting point and merging it with more detailed national scale inventories. The result of such a mosaic inventory still provides a complete dataset but with higher granularity and includes more local knowledge thereby providing modellers with a better starting point for their model exercises which can lead to more accurate (scientific) results and analysis. The merged dataset needs to be carefully evaluated and discrepancies explained and documented as a risk is that "apples and oranges" are treated in the same way and the end result can also be confusion. The advantage of the approach as also stressed by the authors is that the dataset can improve over time incorporating more data as they become available and the current PAPILA dataset is not intended nor expected to be the final dataset but a well-documented starting point that can improve over the years. | We really appreciate your review, it helped us to substantially improve our manuscript, to better communicate our results, and even to detect and correct an inconsistency that we had unintentionally overlooked. We trust that we have responded satisfactorily to all comments in this document. |
| As said such datasets can be an asset for modellers and support / improve regional air quality analysis. The topic is fitting for ESSD and the paper could be published after several clarifications and improvements are made. | |
| My main concerns are on the discussion of the results and clarity of the associated messages. | |
| **Q1:** The idea of the mosaic approach is not new and has been successfully applied in the framework of the Task Force on Hemispheric Transport of Air Pollution (TF HTAP). In the introduction it would be good to refer to the HTAP v2.2 dataset as an example (Maenhout et al., 2015) including the fact that this dataset has been widely used also outside of HTAP. This could be added in the paragraph L68-73 | We appreciate this contribution. We have added the following to the end of that paragraph: "This mosaic approach is an idea that has been successfully applied in the framework of the Task Force on Hemispheric Transport of Air Pollution (HTAP), an international cooperative effort to improve the understanding of the intercontinental transport of air pollution across the Northern Hemisphere. In this context, the HTAP_v2.2 air pollutant grid maps were developed combining available regional information within a complete global dataset (Janssens-Maenhout et al., 2015), and have been widely used even outside of HTAP." |
| **Q2:** The discussion on the impact of the incorporation of the national/local data is unintentionally misleading. For example in L339 "resulting in a difference for SA of only -5%". Strictly speaking this is true BUT you have only replaced 3 countries of total SA and not even for all pollutants (Figure 1). When looking at Table 2 it can be deduced that these 3 countries are only responsible for 20-25% of the total SA emissions (in the case of CO). Thus over 75% of the emissions are unchanged and hence the impact on total SA is very limited. It should be explained in this | We agree with the reviewer and appreciate this observation that helps us improve the way we approach the presentation of results. In the revised version of our manuscript we will restructure the discussion as suggested, for all species. |

| | |
|---|---|
| way and then followed by what the impact is on the three countries that were adjusted. (For CO this is ~-20% which is substantially more than the -5% for all SA, and in fact is still a bit misleading because the countries compensate for each other but this is properly stated in L338. ) This applies to all species (e.g. L365 NOx – same story, this 3% is caused by the bulk of the emission being unchanged because a.o. Brazil and Venezuela are kept constant) | |
| **Q3:**

Modelling is done for MABA (Buenos Aires) only. It should be more clearly stated in the conclusions that these results do not give any insight in the performance for e.g. Colombia, Chile etc. SA is such a big area that a better performance over MABA cannot be taken as sign that also in other countries local data result in better AQ modelling results. I agree it is what we may expect / hope for but it is no prove whatsoever. As an outlook it could be mentioned that an evaluation of the PAPILA dataset over other regions where local data are integrated is not only highly recommended but truly needed to prove the added value of the approach. (see your conclusion section) | This aspect was highlighted by the two reviewers, we agree with them that it is necessary to highlight in the text that a broader evaluation is still needed, evaluating the PAPILA dataset in the other regions where local data were integrated in global datasets. To address that, we will do the following changes:

Lines 255-258 of the original manuscript: the text "The performance of the PAPILA dataset in comparison with that of CAMS as input data to air quality models was assessed using the Weather Research and Forecasting Chemistry (WRF-Chem v4.1.2) regional model. The site chosen for this case study was the MABA, a megacity strongly influenced…", will be replaced by: "The performance of the PAPILA dataset in comparison with CAMS, can be assessed using both inventories as input data of a regional model, implemented in the whole domain where local data has been integrated into the global dataset. This vast region, that includes the tropical Andes in Colombia, the dry Andes in Southern Chile and the Argentinean plateau towards the Atlantic coast, is characterized by a diverse topographic features and vegetation patterns. In order to capture the differences in boundary layer process and surface energy budget in the whole area, a high-resolution model is needed, setup in each area where the main PAPILA/CAMS datasets changes have been made. As a first step of this verification exercise, we present here a study focused on MABA, using the Weather Research and Forecasting Chemistry regional model version 4.1.2 (WRF-Chem v4.1.2). This megacity is strongly influenced…"

In addition, in the conclusions of the revised version of our manuscript we will incorporate a phrase referring to this. |
| **Q4:**

L116 – the shipping might need a bit more discussion. Inland navigation is clear but the domestic coastal (and what do you mean with deep-sea ? ) cannot be so easily separated from SHP-INT – how do you do this? Is that not leading to double counting? SHP-INT will be based on AIS signals but no separation is made between flag states, so how will e.g. Chilean coastal shipping be distinguished from SHP-INT? . | Thank you so much for this comment, because from this we realized that we made a mistake confusing what was actually done with the description and disaggregation of IPCC source categories (not necessarily applicable to an air quality inventory). The splitting in the allocation of emissions from shipping that we actually did in our original work was purely geographical (note that the mosaic made in this work only covers the continental area): we reported under SHP those emissions with spatial assignment within the continental area, and under SHP-INT those assigned outwards from the coast, based on the country and continent masks used for CAMS (CIESIN and CIAT, 2005), consistently with the rest of our work. In this way, the sum of SHP + SHP-INT of our dataset was intended to equal the SHP category of the CAMS. However, from this mistake (for which we apologize) we have introduced an inconsistency only for the case of Colombia, since we built navigation in the PAPILA dataset as follows:

(1) For Chile, as no local data is available, we adopted CAMS data. |

(2) For Argentina, we built PAPILA replacing CAMS with local data only for the continental area, so the emissions outwards from the coast have not been modified. Argentina has local estimates of spatially disaggregated emissions from navigation (Puliafito, et al., 2017), and although work is being done to improve the spatial allocation of emissions from navigation that occur within Argentine territory, the version of the inventory used includes domestic and international navigation. In this way, PAPILA dataset has local information for the continental area of Argentina, and CAMS emissions outwards from the coast.

(3) For Colombia, although our intention was to avoid double counting by removing international bunkers, in this case we omitted what the reviewer correctly points out. In the spatial disaggregation process of this country's emissions, we assigned all domestic navigation within the territory, giving rise to the following inconsistencies: (*i*) an overestimation of the emissions within the territory, since a substantial part of national shipping occurs between the country's offshore maritime ports (and therefore the double counting of them when combined with CAMS outwards from the coast), and (*ii*) the omission of those emissions from international navigation that may occur within the continental area.

To solve all these issues, the following modifications in the manuscript and in the dataset will be made:

(1) To clarify that PAPILA only modifies the continental emissions of CAMS with some local data, but not the emissions located outwards from the coast, the following changes will be included in the text:

(a) The sentence in line 74-75 "This work presents what to our knowledge constitutes the first AEIs from anthropogenic sources covering the entire SA region……", will be replaced by "This work presents what to our knowledge constitutes the first AEIs from anthropogenic sources covering the continental SA region……",

(b) The sentence in line 81-84 "Due to the availability of data in the local AEIs and the completeness of the sectors represented, the 2014-2016 period was selected for this first version of the PAPILA dataset, including local information from Argentina (Puliafito et al., 2017; Castesana et al., 2018), Chile (Mazzeo et al., 2018; Gallardo et al., 2018) and Colombia (IDEAM, 2017)." will be replaced by "Due to the availability of data in the local AEIs and the completeness of the sectors represented, the 2014-2016 period was selected for this first version of the PAPILA dataset, including local information from de continental areas of Argentina (Puliafito et al., 2017; Castesana et al., 2018), Chile (Mazzeo et al., 2018; Gallardo et al., 2018) and Colombia (IDEAM, 2017)."

(2) We will modify the PAPILA dataset as follows:

(a) Replacing local data from SHP in the current version of the dataset Colombia with CAMS data, as in the case of Chile.

(b) Presenting the emissions from shipping in an aggregated way in line with CAMS approach. To such end, we will aggregate the originally called SHP and SHP-INT in a unique category, that will be called SHP = (SHP + SHP-INT) (as CAMS does).

This will lead to a new dataset version that will be uploaded to the Mendeley data repository. The final product resulting from the full review process will be published as http://dx.doi.org/10.17632/btf2mz4fhf.3 instead of http://dx.doi.org/10.17632/btf2mz4fhf.2.

(3)  We will add into the text a clearer description of our approach to build PAPILA's dataset for this category, as follows:

(a)  The paragraph in lines 116-117 "inland navigation, which includes domestic coastal, deep-sea and inland waterborne navigation, (SHP); international navigation (SHP-INT)" will be replaced by "domestic and international navigation (SHP)".

(b)  The paragraph in lines 119-120 "International navigation emissions were taken entirely from the CAMS database", will be removed, since in Figure 1 it will be seen that only Argentina includes local data from shipping (Chile and Colombia will have the CAMS data), and the procedure will be detailed in the subsection 2.2.1 of the revised version of the manuscript.

(c)  The paragraph in lines 152-153 "(*vi*) for inland navigation, fuel consumption, spatially distributed with the geographical identification of the berths routs and ports boundaries" will be replaced by "(*vi*) for inland navigation (namely, domestic plus international navigation on the continental area of Argentina), spatially distributed with the geographical identification of the berths routes and ports boundaries"

(d)  At the end of the sentence in lines 162-164: "Final emissions were adapted to a homogeneous grid of 0.1° x 0.1°, and combined with agricultural local inventories described below, and with the CAMS information on emissions from SWD", we will add "and from SHP outwards from the Argentine coast".

(e)  The phrase in lines 344-348: "By downscaling B. Blanca urban domain we identified the absence of emissions from shipping activities (inland: SHP, and international: SHP-INT) in the global inventory. While emissions from SHP were estimated locally, estimates for SHP-INT were not available and therefore they were taken from global estimates. In this domain, international navigation is a concern…" will be replaced by ""By downscaling B. Blanca urban domain we identified the absence of emissions from shipping activities in the global inventory. While emissions from SHP within the continental area were estimated locally, offshore emissions were taken from CAMS, which reports zero emissions for this region. In this domain, emissions from navigation activities are a concern…".

(f)  The phrase in lines 529-530: "(*ii*) domestic and international waterborne navigation are not reflected by downscaling the port city B. Blanca in Argentina" will be replaced by "(*ii*) navigation activities are not reflected by downscaling the port city B. Blanca in Argentina".

(4)  We will modify the tables and figures where we have made use of the notation for shipping, unifying the use of the new SHP category (Figure 1, Table A1).

(5)  The changes in the estimates for the emissions from SHP in Colombia will be reflected in Figure 3 and 4, in Table 2, as well as

| | in the total amounts of CO, NOx and SO2 emissions in its territory, which will be incorporated the corresponding subsections of the discussion in the revised version of our manuscript. |
| --- | --- |
| | (6) In the rest of the manuscript, further changes can be included to be consistent with the major changes made in SHP. |
| Q5:

L398. The main point is that CAMS treats countries uniform w/o correcting for the climatic zones in the country. However, SA countries are large and the climatic / temperature zones vary widely from tropics to antartica. In the discussion this could be mentioned and suggested as an improvement for global inventories as the temperature zones are well known and available on grids. This could be used to redistribute within a country. | Thanks for this contribution. We have included the following phrase in section 3.3 (PAPILA-CAMS main differences): "It is also worth mentioning that, unlike local inventories, CAMS treats countries uniformly without correcting for the climatic zones, which vary widely within many of the SA countries". In our answer to Q7, this phrase is seen in context. |
| Q6:

L504 – here and throughout the MS – where you use "driver" you mean "proxy". (Economic growth is a driver for growing emissions; population density is a proxy to distribute such emissions). Please check the whole section 3.3. and further for this terminology. | Modified as suggested. |
| Q7:

The discussion in L505-511 is unclear, please rewrite and check for using the right words. (it can be a low-resolution map but not low-resolution density) | In response to this observation, and also to Q5 and Q48, the paragraph indicated has been modified as follows: "Another relevant aspect of national and therefore international statistics is the lack of reliable information on firewood consumption, widely used in rural areas of SA and even in some urban areas, such as the cold regions of Chile. This fact also impacts on the correct representation of the replacement of firewood by LPG or Natural Gas that took place in Argentina in the last decade, due to higher production of non-conventional shale gas in Vaca Muerta basin (El Pais, 2015) and the resulting reduction in fossil fuel prices. Additional differences between local and global datasets are related with the different resolutions of the population distribution maps used as proxies for the spatial distribution of the emissions from some categories. Local inventories use population density information based on higher resolution maps than those used by the global ones. This is clearly noticeable not only in the local-global differences found by downscaling urban domains, but also in the spatial coverage of RES emissions in global inventories, where emissions are assigned even to large non-populated areas, such as the Amazon rainforest or some desert areas of the region (Figure 4). It is also worth mentioning that, unlike local inventories, CAMS treats countries uniformly without correcting for the climatic zones, which vary widely within many of the SA countries. Broader discussions on emissions from RES are given by Puliafito et al. (2021) and Álamos et al. (this issue) for Argentina and Chile, respectively". |
| Q8:

L512 as should be or? ; population as a driver should be "population density as a proxy" | In this case, the term "driver" is correct since we are talking about projection of emissions and not of proxies for their distribution. As mentioned by Hoesly et al., 2018, "Activity drivers for non-combustion sectors in modern years are primarily population estimates". It is true that the sentence is confusing, and we decided to modify it as follows: "For non-combustion sources, as many industrial processes, population |

| | estimates are used as drivers for the CAMS projections (based on CEDS trends) (Hoesly et al., 2018)". |
|---|---|
| Q9:

L562 remove "which is reflected in the uneven levels of development of local inventories" as such it does not add information. | Modified as suggested. |
| Q10:

Final paragraph – I think you can add that especially Brazil and 1 or 2 others should be added because the 3 countries added now are responsible for less than 25% of the emissions. How would this change if Brazil is add? | In the conclusions of the revised version of our manuscript we will incorporate a phrase referring to what is indicated here. |
| Q11:

L571 sentence "in addition etc. can be removed, the sentence before is a stronger ending with a clear message. | Modified as suggested. |
| Q12:

Table A2 – is that really a summary of NOx ? or do you mean NH3? I'm surprised to see NOx emissions come from dairy cattle. | We have checked that the content on $NO_x$ emissions in Table A2 is correct. As described in Chapter 3.B Manure management of the EMEP/EEA air pollutant emission inventory Guidebook 2016 and 2019, NO is formed initially through nitrification and subsequently also by denitrification in the surface layers of stored manure or in manure aerated to reduce odour or to promote composting. NO emissions from soils are generally considered to be products of nitrification. Increased nitrification is likely to occur after the application of manures and the deposition of excreta during grazing. NO emissions arising from livestock housing and manure stores should be reported under NFR 3B (manure management), while those arising after the application of manures to land or from grazed pastures should be reported under NFR 3D (agricultural soils). Following these guidelines, in our work we have reported emissions from managed excreta as AGL, and those deposited in pasture during animal grazing under AGS (lines 177-178 of the original manuscript), for all livestock. According to Castesana et al. (2018), dairy cattle in Argentina is disaggregated into two sub-classes: (*i*) dairy cows and (*ii*) other dairy cattle composed of animals for womb reposition and reproducers. Dairy cows spend ten months in production (lactating cows) and two months resting (dry cows). Lactating cows spend ~2 h a day in milk rooms (whose excretions are estimated and reported as manure management) and the remaining time in the field together with dry cows and other dairy cattle whose excretions are deposited on pasture. It is true that, compared to other livestock, the $NO_x$ emissions of dairy cattle are not relevant, and it does not seem to make much sense to present them in a disaggregated manner in Table A2. However, we did it this way because it is information of interest at the national and regional level.

Nevertheless, from this observation, when checking the content of the table we have found that there was an error in the transcription of NMVOCs emissions of non-dairy cattle (both for AGL and AGL), for which in the original manuscript we repeated the results of 2016 in other years. We have corrected it as follows: |

| NMVOCs (Gg.y-1) | | | 2014 | 2015 | 2016 |
|---|---|---|---|---|---|
| AGL | Manure management | Dairy cattle | 0,62 | 0,62 | 0,63 |
| | | Non-dairy cattle | 1,27 | 1,30 | 1,32 |
| | | Other livestock | 11,04 | 11,08 | 10,92 |
| | | Total from AGL | 12,93 | 13,00 | 12,86 |
| AGS | Manure in pasture | Dairy cattle | 1,25 | 1,25 | 1,25 |
| | | Non-dairy cattle | 12,82 | 12,93 | 13,01 |
| | | Other livestock | 0,10 | 0,11 | 0,11 |
| | | Total from animal grazing | 14,17 | 14,29 | 14,37 |
| | Fertilizers | | NO | NO | NO |
| | Crops | | 29,88 | 34,23 | 33,73 |
| | Total from AGS | | 44,05 | 48,52 | 48,10 |

| | |
|---|---|
| **Minor issues:** | |
| Q13: L5 place "derived" after "data" | Modified as suggested. |
| Q14: L13 "obtained" should be "found" | Modified as suggested. |
| Q15: L15 ….lower levels…. | Modified as suggested. |
| Q16: L20 replace: PAPILA-based modelling results had a lower bias for CO and NOx concentrations in winter while CAMS-based results for the same period tended to deliver an underestimation of these concentrations. | The original sentence "For winter, PAPILA-based results had lower bias for CO and $NO_x$ concentrations, for which CAMS-based results tended to be underestimated" was replaced by "PAPILA-based modelling results had a lower bias for CO and $NO_x$ concentrations in winter while CAMS-based results for the same period tended to deliver an underestimation of these concentrations". |
| Q17: L28 From the energy standpoint -> Regarding energy use, | Modified as suggested. |
| Q18: L39 it's a bit awkward to use a reference of 20 years ago here. Much would / may have hanged since then. I think this is also discussed in Huneeus et al. 2020; might be a added here? | Modified as suggested. |
| Q19: L43 change larger to increasing | Modified as suggested. |
| Q20: L77 change bibliography to literature | Modified as suggested. |
| Q21: L91 choose either LAC or SA and use only one. Please check full MS and SI for this | The acronym LAC was removed as it is not mentioned in the rest of the manuscript. Instead we have written "Latin America and the Caribbean", since in that paragraph we refer to the objectives of the PAPILA project that apply to that region. In the rest of the document we use SA. |
| Q22: L98 2 times have | Modified as suggested. |
| Q23: L99 Explain a bit better why you need complete country inventories (because this is the common / shared administrative entity that can be exchanged with the global inventory) | We appreciate this observation. The phrase "For this reason, the information from these countries was not included in this first version of the combined dataset" in the original manuscript, was replaced by "Since national territories are the common administrative entity that can be exchanged with the global inventory, the local information on emissions from these countries was not included in this first version of the combined dataset". |

| | |
|---|---|
| **Q24:**

L140 – yes that is a solution but it is not really necessary, the combination of coordinates AND a country code could also generate unique values and the cell can that be shared by 2 or even 3 countries. The difficulty is probably that the global inventory does not have these country codes | We agree with this comment. However, to be consistent with the base inventory, we decided to use the country and continent masks applied by CAMS (CIESIN and CIAT, 2005), in which each cell is assigned a unique country code (they are not shared by 2 or more countries). For clarification purposes, we have replaced the phrase "This problem was solved using the country and continent masks created at 0.1° resolution (CIESIN and CIAT, 2005) that assign a unique value for each cell" in the original manuscript, by "To be consistent with the base inventory used in this work, this problem was solved using the country and continent masks applied by CAMS (CIESIN and CIAT, 2005), which are created at 0.1° resolution assigning a unique country value for each cell". |
| **Q25:**

Figure 1 – I would call that a Table and not a figure. Replace SA on the bottom with "Rest of SA" or "Other SA" | We replace SA with Rest of SA as suggested. Regarding the Table/Figure, it is true that it looks like a double entry table, but we decided to name it Figure because the journal's format criteria do not allow Tables in colour or in pdf or image format, and we thought it was simpler and clearer to present it this way. |
| **Q26:**

L156 explain how you distributed the rest as area sources? With what proxy? | The paragraph in the original manuscript: "The GEAA inventory has been updated for this work including emissions from IND, which were not covered in the published (Puliafito et al., 2017). With these changes for manufacturing industries, the dataset considers fuel consumption by fuels, petroleum refining and emissions from the production process itself for the main industries, spatially distributed with the location of the main industries and distributing the rest as area sources in the whole territory. In all these categories the combustion of fossil and biomass fuels was considered.", was replaced by: "The GEAA inventory has been updated for this work including emissions from IND, which were not covered in the published (Puliafito et al., 2017). These emissions include (*i*) those from fuel consumption and from production process itself for the main industries, disaggregated by fuel and spatially distributed with the precise location of each facility, and (*ii*) those from fuel consumption of small industries, whose consumption is known by activity and by district, and whose spatial disaggregation of emissions was carried out using the population density of each district as a proxy". |
| **Q27:**

L179 taken to -> converted to | Modified as suggested. |
| **Q28:**

L183 Such => This | Modified as suggested. |
| **Q29:**

L191 considered => assumed | Modified as suggested. |
| **Q30:**

L215 check word missing (we to report?) | The sentence "Given that the local inventory reports ENE and IND (including use of solvent) emissions together and that insufficient information foro spatial disaggregation was available, we to report ENE + IND under the IND sector for the case of Chile" was rephrased as "Given that both the magnitudes and the spatial distribution of emissions from ENE and IND (including use of solvent) are reported in an aggregate way in the Chilean inventory, we decided to report them under the IND category". In this way, the indicated mistake is corrected. |

| | |
|---|---|
| Q31:

L228 applied => apply | Modified as suggested. |
| Q32:

L378 minority => minor | Modified as suggested. |
| Q33:

Table 2. it would be good to indicate that the first 3 subregions are in Argentina and the last 3 in Chile. This is well explained in the text but for international readers it might still be difficult to remember this. Moreover it should be considered (but only a suggestion) if it is not better to e.g. in L250 where MABA is introduced to simply state that in this study Buenos Aires includes the entire Metropolitan area of BA. In the paper you can then discuss simply Buenos Aires and do not have to use this MABA which for many people will be something they are not familiar with. Same applies to MRS. Everybody knows Santiago is the capital of Chile, MRS will not be known to many. Also in the Table 2 acronyms like MABA and MRS are not very helpful. You could add a table footnote that the entire metropolitan region is included and use the city names in the table. | The table was modified as suggested. Although "Buenos Aires" is an Argentine province and does not strictly correspond to MABA, and "Santiago" is the name of the capital city and does not correspond to MRS either, we agree with the reviewer that the acronyms MABA and MRS are not very reader friendly. We accepted the reviewer's suggestion, clarifying that Buenos Aires will refer to the MABA and Santiago to the MRS in the suggested paragraph. The acronyms MABA and MRS were replaced by Buenos Aires and Santiago, respectively, throughout the manuscript and in the corresponding tables and figures. |
| Q34:

Table 2 – I suggest to only show a digit if the value is smaller than 100. E.g. 31225.2 is not very helpful. PAPILA NOx column has 2 times 16.8 – pls check if correct | Modified as suggested.

The data in the $NO_x$ column have been checked and the values are correct: Bahía Blanca 16.81 and Mendoza 16.84. |
| Q35:

Fig 3 . Pls consider if it is not better to change the order and show Argentina – followed by its 3 sub regions (same for Chile). This may help to see patterns. For example when looking at SO2 the pattern for ARG and sub region is exactly the same (so it's a national consistent difference) for Chile it is different with a redistribution between Antofagasta and MRS. | Modified as suggested. |
| Q36:

L415. How relevant are these emissions? NH3 is dominated by AGRI; 80% sounds big but if 80% of almost nothing is still nothing. Is it worth the attention? | Although in the last sentence of the paragraph we added that "as in the case of urban domains in Argentina, the contribution of each domain to the total emissions in Chile was negligible", we agree that the content of this paragraph is not worth the attention. We will take this observation into account when restructuring the discussion (as suggested in Q2) in the revised version of the manuscript. |
| Q37:

L430 non => not | Modified as suggested. |
| Q38:

L435 see earlier remark on comparison for the entire region (which is not informative as bulk of the emissions are taken from CAMS and thus constant) | Noted. |
| Q39:

Table 3 – explain acronyms in table footnote (NMB, NMGE, FAC2) | Modified as suggested. |

| | |
|---|---|
| Q40:

L457 null => no | Modified as suggested. |
| Q41:

L460 goodness of the => better | The sentence "Thus, the goodness of the PAPILA-based results exhibited for winter were not that apparent for summer" in the original manuscript was replaced by "Thus, the results for the summer simulations were not as conclusive as winter simulations". |
| Q42:

L467 high => height | Modified as suggested. |
| Q43:

Fig 6 change top legend : remove "simul" (it is not a known word), make it bigger and have e.g. CAMS simulation; PAPILA simulation | Modified as suggested. |
| Q44: L477 goodness of => quality | Modified as suggested. |
| Q45:

L491 than => as | Modified as suggested. |
| Q46:

L495 what is intended with hydraulic availability??? Change and explain better, different word? | The original phrase "...which in turn are strongly correlated with hydraulic availability not captured by the extrapolations" was replaced by "...which in turn are strongly correlated with the amount of water available for hydropower generation, not captured by the extrapolations". |
| Q47:

L497 the incorporation of diesel-based motor generators => the use of diesel-fuelled generators | Modified as suggested. |
| Q48:

L504-505 Why does this imply a difficulty? | We have modified this sentence. Please see the answer to Q7. |
| Q49:

L520 tools => monitoring | Modified as suggested. |
| Q50:

L520-527 – Please rewrite this paragraph and check with co-authors; it is rather cryptic. E.g. "information generated from a local perspective" can simply be "local information" | As suggested, we will modify this paragraph by writing it more concisely, check it with co-authors, and include it in the revised version of our manuscript. |
| Q51:

L529 Why are aviation emissions not included? Are they not important? Not provided by EDGAR or CEDS? | The global databases EDGARv4.3.2 and CEDS include emissions from aviation, as well as the regional versions of CAMS (CAMS-REG-GHG and CAMS-REG-AP) on European territory. However, and unlike the datasets mentioned, what we have found and as we mentioned in the indicated line is that the version of the global inventory CAMS-GLOB-ANT v4.1 (used as a base inventory in this work) omits this category. Based on this observation, we have decided to add the following sentence to subsection 2.1 PAPILA dataset overview of our manuscript: "To be consistent with the base inventory (CAMS-GLOB-ANT v4.1) used for our mosaic inventory, aviation emissions were not included in this first version of the PAPILA dataset". Additionally, in the revised version of our manuscript we will include the aforementioned in the paragraph indicated by the reviewer. |
| Q52: | |

| | |
|---|---|
| L534 item v) – Please explain this better and be a bit more informative on its relevance – are these emissions missing or possibly reported elsewhere? If it is CO, NOX, SO2 it must be from waste burning not landfills or waste water. Is this not present for all SA counties? Or only for Chile and Columbia? When you look at the local information where you have these emissions included, how important is it? What is the share for the country or city? If it is missing but only adds e.g. 0.1% to the national total is it relevant? | We found out that CAMS assigns zero emissions of CO, $NO_x$ and $SO_2$ from SWD for all SA, except for Brazil and the Guianas. The annual emissions from SWD of these species are not relevant at the national level, however, since we are aware that they come from burning and are therefore generated during short periods and in small areas (a situation that deserves attention in air quality studies), we do not want to fail to mention this gap in the global inventory. We are also aware that SWD emissions (not only waste burning) are difficult to represent in our region, both for global and for local inventory developers, and we refer to this with the phrase of the lines 538-540 of the original manuscript. From this observation, we will better explain this aspect and include the aforementioned in the revised version of our manuscript, also incorporating in this paragraph the Q4 and Q51 observations of the reviewer and our respective comments. |
| Q53:

L541 replicated in => addressed in the | Modified as suggested. |
| Q54:

L541 remove "for the case of" | Modified as suggested. |
| Q55:

Section 4 – I did not check the dataset and the sums with the tables. Please ask one of your 10 co-authors who was not involved in uploading the data to download and do a check. I assume it will be good but advise to do this check. | We will check our new version of the PAPILA dataset (with corrected SHP) before sending the revised version of the manuscript. |
| Q56:

L552 made for a limited number of countries – change to "only for Argentina, Chile and Colombia" (be as specific as possible) and add that this is a living dataset and that in the future other countries can be added. | We agree with the reviewer to be as specific as possible, and we liked the concept of "living inventory". These suggestions will be incorporated when restructuring the conclusions of the revised version of the manuscript. |